# Transcription modulates chromatin dynamics and locus configuration sampling

Giada Forte[1], Adam Buckle[2], Shelagh Boyle[2], Davide Marenduzzo [1],
Nick Gilbert [2] ✉ & Chris A. Brackley [1] ✉

In living cells, the 3D structure of gene loci is dynamic, but this is not revealed by 3C and FISH experiments in fixed samples, leaving a notable gap in our understanding. To overcome these limitations, we applied the highly predictive heteromorphic polymer (HiP-HoP) model to determine chromatin fiber mobility at the *Pax6* locus in three mouse cell lines with different transcription states. While transcriptional activity minimally affects movement of 40-kbp regions, we observed that motion of smaller 1-kbp regions depends strongly on local disruption to chromatin fiber structure marked by H3K27 acetylation. This also substantially influenced locus configuration dynamics by modulating protein-mediated promoter-enhancer loops. Importantly, these simulations indicate that chromatin dynamics are sufficiently fast to sample all possible locus conformations within minutes, generating wide dynamic variability within single cells. This combination of simulation and experimental validation provides insight into how transcriptional activity influences chromatin structure and gene dynamics.

The spatial organization of the chromosome around a gene is thought to be intimately linked to its expression, providing an important mechanism for gene regulation. At short length scales (<5 kilobase pairs (kbp)), this mechanism might involve nucleosome repositioning and disruption of the chromatin fiber structure[1,2], modifying the accessibility of the DNA to proteins. At larger length scales (>5 kbp), this could involve the way chromatin forms loops bringing together promoters and their enhancers. Advances in microscopy[3,4], and in next-generation sequencing methods such as chromosome-conformation-capture (3C) and its variants[5], have revealed much about gene locus structure. More recently, the development of structural probes operating at the single-cell level has revealed striking variability within a tissue or population of cells that are phenotypically homogeneous[3,4,6]. The compatibility of such cell-to-cell variability with a robust transcriptional program and emerging phenotype is remarkable and still awaits full explanation. The variability of gene loci was exemplified by our recent modeling work on the locus of *Pax6*, a highly regulated and highly conserved developmental gene. Our simulations showed that chromatin interaction patterns revealed by CaptureC experiments could be generated by locus conformations that vary widely from cell to cell, and that the level of variation (validated using DNA fluorescence in situ hybridization microscopy (FISH)) also markedly depends on cell type and expression level[7].

Although experimental methods probing chromosome organization continue to improve, the majority of studies to date have focused on fixed cells, providing information on only a 'snapshot' in time. An understanding of how locus conformations evolve dynamically remains largely elusive. For example, it is unclear whether the structural variation observed across a population is representative of the configurations adopted dynamically within a single cell. Or, does the chromatin in a single cell only visit, or sample, a small part of this 'configuration space,' with the observed variability arising only when gathering data from many cells? In other words, if one were to track locus configuration in a live cell, would one observe wide variations or a relatively static picture? The answer to this may give insight into how such variability can still give tight control of expression and phenotype. Although live-cell imaging has advanced markedly in recent years, challenges remain. Super-resolution microscopy allows ever-increasing spatial

[1]SUPA, School of Physics and Astronomy, University of Edinburgh, Edinburgh, UK. [2]MRC Human Genetics Unit, Institute of Genetics & Cancer, University of Edinburgh, Western General Hospital, Edinburgh, UK. ✉e-mail: Nick.Gilbert@ed.ac.uk; C.Brackley@ed.ac.uk

resolution to be achieved, and high-throughput techniques, such as Hi-D[8], have been developed to monitor chromatin diffusivity in vivo. However, it remains difficult to reach high temporal resolution while labeling multiple points of interest simultaneously. Here, we use biophysical modeling and computer simulations[9] to study and predict dynamics at the *Pax6* locus. This approach provides mechanistic insight as well as new hypotheses and testable predictions, which we hope will stimulate further experiments.

Previously, we developed the HiP-HoP simulation framework to predict structural information on a gene locus at both the single-cell and population level. This was applied to several gene loci, including those of *Pax6* in mouse, *SOX2* (ref. 7) and *CCND1* (ref. 10) in human, and (using an earlier version) mouse alpha- and beta-globin[11]. Here, we evolved HiP-HoP such that it can be used to analyze the dynamics of the *Pax6* locus in three mouse tissue-derived cell lines that express the gene at different levels[12,13]. We first examined the dynamic properties of simulated chromatin regions at different length scales and found that these vary substantially across the locus. The mobility of a given segment depends both on its biophysical properties (protein binding, local chromatin compaction) and on its surroundings, with local macromolecular crowding playing a pivotal role (we use the term 'mobility' to refer to a measure of how far a given segment will move or diffuse within a given time). We then studied the dynamics of the overall locus configuration, analyzing the timescale over which promoter-enhancer interactions change. The results suggest that the locus can sample all of its different configurations within minutes, strongly indicating that wide dynamic variation of structure would be observed within a single cell. Finally, we performed interventional experiments, in which we gave cells treatments either to inhibit transcription or to release topological strain; we made measurements in fixed cells and compared these with dynamic simulations. Interestingly, we found that removing proteins representing polymerase complexes from the simulation did not reproduce the changes observed in the transcription inhibition experiment; this led us to an alternative model scenario.

## Results

### Modified HiP-HoP framework to explore locus dynamics

We have previously used HiP-HoP[7,9,10] to study the structure of the *Pax6* locus in three mouse cell lines, denoted *Pax6* OFF, ON, and HIGH (indicating the expression state of the gene). The model represents a chromosome region as a chain of beads and combines three key mechanisms that organize the locus (Fig. 1a). First, model proteins (representing complexes of RNA polymerase and transcription factors) that diffuse within the system can interact with the chromatin at specific binding sites; importantly, these proteins are multivalent and can bind multiple sites simultaneously, forming molecular bridges between distal chromatin sites[14,15]. Second, the loop extrusion mechanism is included[16–18]; this asserts that the cohesin complex can actively push chromatin into loops, which grow until they are stabilized by CTCF proteins bound in a convergent orientation[19]. Finally, the model includes a 'heteromorphic polymer' description of chromatin, meaning that the biophysical properties, for example those determined by the fiber structure, can vary along its length[1,2,20]. For this, we discriminate between two chromatin states, one with a more compact, thicker fiber, and one with a thinner, less compact (that is, more open, disrupted, or flexible) fiber[21] (see Methods and Supplementary Notes). Here, we predominantly consider active chromatin regions, but this model could be extended to account for repressed genomic regions[10,22]. An important feature is that, in the simulations, proteins tend to come together into clusters; this is driven by a mechanism known as the 'bridging induced attraction'[15,23] and is a consequence of a protein's ability to form molecular bridges between distal chromatin sites. Clusters form around two or more protein-binding sites on the chromatin and represent the foci or 'phase-separated droplets' of transcription-associated proteins observed in recent microscopy studies[24,25] (although the

mechanism through which this occurs in vivo is not fully understood). In HiP-HoP, proteins continually switch between a chromatin-binding and a non-binding state[26], modeling post-translational modifications and resulting in clusters having realistic protein dynamics in terms of the exchange between the cluster and the soluble pool.

Three data sets are used as an input to HiP-HoP. First, DNA accessibility data (assay for transposase-accessible chromatin with sequencing (ATAC-seq)) are used to identify protein-binding sites (we use a simplifying assumption that ATAC peaks coincide with binding sites for 'active' proteins). Second, chromatin immunoprecipitation (ChIP) data on acetylation of lysine 27 on histone H3 (H3K27ac) are used to identify regions that are in the more open chromatin state: this decision is based on previous studies suggesting that chromatin possessing this mark has a disrupted structure[2,27]. Finally, ChIP data on CTCF and cohesin (Rad21) are used to identify positions of loop-stabilizing anchors (see Methods and Supplementary Notes).

In our previous study of *Pax6*, we used HiP-HoP to generate a population of simulated locus structures, which provided good predictions of experimental measures at both the population (validated by CaptureC) and single-cell (validated by FISH) level. CaptureC is a many-to-all 3C method which probes genome-wide interactions at a set of selected target locations[28,29].

Obtaining realistic dynamics of locus structure required substantial changes to the simulation set up. We hypothesized that local variation in chromatin density might play a role in variation of dynamics across the simulated region, so we therefore used a more realistic overall chromatin density and simulated a larger chromatin fragment than in previous work. For computational efficiency, a 3-mega-base-pair (Mbp) region around *Pax6* (chr2:104,000,000–107,000,000 mm9 genome build) was selected and concatemerized 10 times along a 40-Mbp fiber (that is, one simulation is equivalent to taking measurements of the locus across 10 single cells, Fig. 1c,d). Simulations representing dynamics in 20 single cells from each cell line were performed, and we determined that extracting 400 configurations at regular time intervals provided a good representation of locus motion (see Methods; example configurations are shown in Fig. 1e).

To ensure that the simulations still gave good predictions of locus conformation with this new set up, results were compared with both CaptureC data (Extended Data Fig. 1) and FISH (Extended Data Fig. 2). CaptureC targets were positioned at promoters and CTCF-binding sites across the locus (Fig. 1b, green stars). Three FISH probes were selected to cover the *Pax6* promoters and two previously identified enhancers, denoted the upstream and downstream regulatory regions (URR and DRR; Fig. 1b, purple blocks), and were used in three-color imaging experiments to obtain simultaneous measurements of probe separations in single cells. Two quantitative metrics (see Supplementary Notes) confirmed that conformations predicted by this new version of HiP-HoP showed similar levels of agreement with experiments as the original model[7] (Extended Data Fig. 3).

### Dynamics depend on transcription and fiber structure

To explore the dynamics around *Pax6*, we used a simulation strategy analogous to live-cell tracking of specific regions of the locus corresponding to the positions of the *Pax6*, URR, and DRR FISH probes (Fig. 2a). The mean squared displacement (MSD) was calculated as a function of lag time for each probe (Fig. 2b–d). In these simulations, the length-scale of the chromatin (the diameter of the bead ($\sigma$)) is 17.6 nm, determined by comparing simulated and experimental FISH measurements. The simulation time unit ($\tau$) is approximately 2.07 ms, determined by comparison with previous motion-tracking experiments[30]. This gives, for each simulation, an approximate total duration of 27 min (see Methods and Supplementary Notes for details on this mapping).

MSD curves for different probes, and for a given probe in different cell lines look highly similar, indicating that the mobility of these regions is under similar constraints. For a freely diffusing object, the

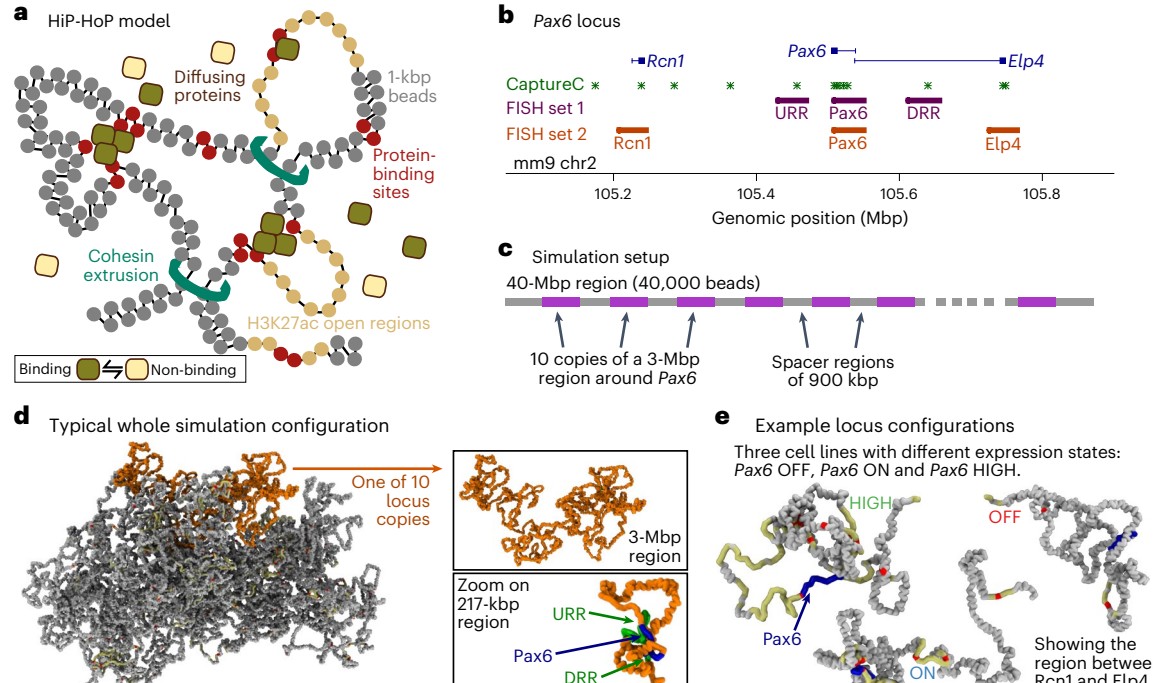

**Fig. 1 | Model schematic and simulation set-up. a**, In the HiP-HoP model, a chromosome region is represented by a bead-and-spring polymer, in which each bead represents 1 kbp of chromatin. The model includes diffusing protein complexes represented by spheres, loop extruders represented by additional springs, and a heteromorphic structure in which some polymer regions have additional next-nearest neighbor springs that lead to local crumpling of the chain. Proteins stochastically switch between a binding and a non-binding state; when in the binding state, there is an attractive interaction between the proteins and their binding sites on the chromatin. Further details are given in the Methods and Supplementary Notes. **b**, A map of the mouse *Pax6* locus (mm9 genome build). Positions of CaptureC targets or viewpoints and two sets of FISH probes are indicated (see Supplementary Tables 1 and 2, respectively, for genomic coordinates). **c**, Schematic showing the simulation set-up. Simulations of a 40-Mbp chromatin fiber (40,000-bead polymer) were performed. Ten copies

of a 3-Mbp region around the *Pax6* locus were included on each fiber, allowing multiple results to be obtained from each simulation. Copies of the locus used input data from one of the three cell lines; in repeat simulations, versions of the locus from different cell lines were in different positions. In this way, each locus experienced a similar surrounding environment. **d**, Simulation snapshot showing a typical configuration of the 40-Mbp fiber. One of the ten copies of the locus is shown in orange. In the top right, this same copy of the locus is depicted with the rest of the fiber not visible. In the bottom right, a zoom shows only the region immediately surrounding *Pax6* and two nearby regulatory regions (upstream and downstream regulatory regions, URR and DRR). **e**, Further simulation snapshots show example configurations of the locus (only the ~500-kbp region between *Rcn1* and *Elp4* is shown) in each of three cell lines in which *Pax6* is expressed at different levels. Chromatin regions with H3K27ac are shown in yellow, and ATAC-seq peaks (binding sites) in red. The *Pax6* gene body is shown in blue.

MSD would grow linearly with time, but as expected for motion of a polymer segment, the MSDs here grew more slowly[31] (inset Fig. 2b–d). Some slight differences between cell lines were observed at large times (e.g., longer than 600s); particularly, for the *Pax6* probe the MSD grew more slowly for *Pax6* ON cells than for the others. Considering the input data, which determine the chromatin properties, *Pax6* ON cells have more protein-binding sites (ATAC-seq peaks) than the other two cell lines, suggesting that protein-mediated chromatin looping may lead to reduced mobility.

To further quantify the dynamics, we defined a mobility measure (*M*) as the MSD after a fixed lag time of $10^4 \times \tau \approx 20.7$ s. For each cell line, *M* was calculated for each 1-kbp bead to obtain a 'mobility profile' over the locus (Fig. 2e). These profiles revealed large differences in mobility both across the locus and between cell lines. Most strikingly, around the DRR, the mobility of *Pax6* HIGH cells was around 1.5 times higher than that of ON or OFF cells. In general, mobility tended to be higher in regions with H3K27ac (Fig. 2e, yellow blocks) and lower at ATAC-seq peaks (red blocks). This is in stark contrast to the FISH probe MSDs, for which little difference between probes or cell lines was observed. This shows that dynamic measurements are highly sensitive to the way that chromatin is probed: local variation in mobility of different 1-kbp beads was largely obscured when we analyzed ~40-kbp probes, suggesting that the former depends on the

behavior of chromatin over small length scales. Simulating probes of different sizes revealed that both the observed mean and s.d. of mobilities across the locus decreased with increasing probe size (Extended Data Fig. 4a,b).

Measurements of local macromolecular density around each polymer bead (Fig. 2f) revealed that this has a strong effect on diffusion. A lower local density was observed in regions that were more mobile (Fig. 2e): there was a significant negative correlation (Pearson correlation between −0.5 and −0.6, $P < 10^{-10}$; Fig. 3a). It is also clear that the local density was enriched at protein-binding sites, visualized by the color coding of points in Figure 3a. Beads at ATAC-seq sites (red) had a 5.15% lower mobility, on average, than that of other beads. Reducing the number of model proteins in simulations led to a reduction in local densities and an increase in mobility at binding sites (Extended Data Fig. 4c,d), confirming that protein binding and bridge formation impair motion. By contrast, the mobility of H3K27ac-marked beads (yellow in Fig. 3a) was 7.13% higher, on average, than that of other beads. This is consistent with the more compact (non-H3K27ac) regions being more constrained than H3K27ac regions within the fiber. Increased mobility has previously been associated with histone acetylation in live-cell imaging[32]; in accordance, there was a positive correlation between the mobility of 20-kbp windows and H3K27ac density in our simulations (Fig. 3b; Pearson's $r = 0.43$, $P < 10^{-10}$).

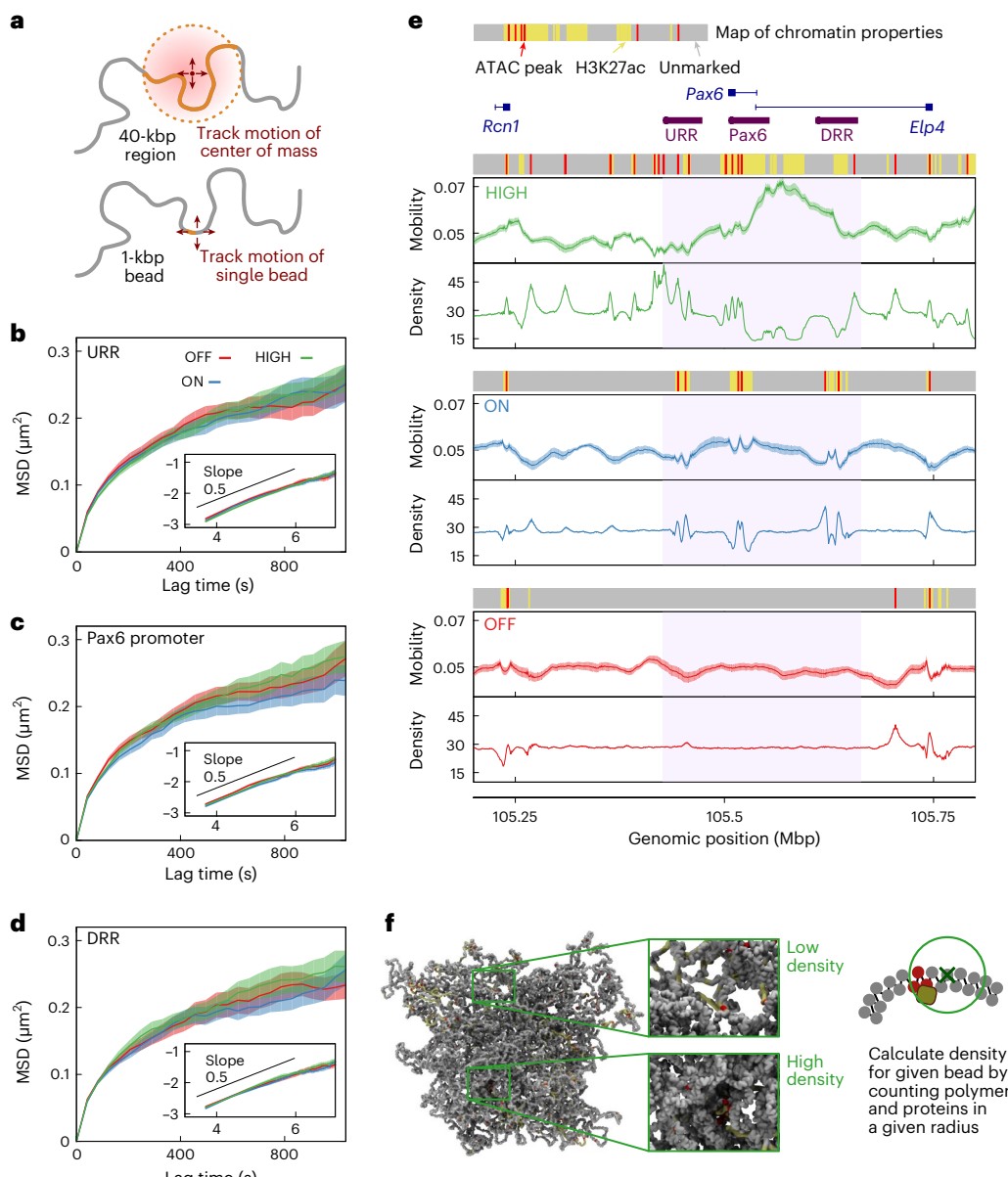

**Fig. 2 | Dynamics of individual chromatin regions. a**, To quantify the dynamics of a region covered by a FISH probe, the MSD of the center of mass of that region was computed (top). Alternatively, the MSD of a single chromatin bead was computed (bottom). **b–d**, Plots showing the MSD of simulated fluorescent probes positioned at the *Pax6* promoter (**c**) and the two distal regulatory regions (URR (**b**) and DRR (**d**)). These cover the same regions as the FISH probes used in fixed-cell experiments (probe set 1 in Fig. 1b). Simulation lengths and times were mapped to real units, as detailed in the Methods and Supplementary Notes. All results were obtained from 20 independent simulations of the locus; results from three cell lines are shown (lines) and the shaded regions indicate the s.e.m. The inset plots show log(MSD) *vs.* log(lag time), and the black lines have a slope of 0.5. **e**, For each cell line, chromatin mobility is plotted as a function of position

across the locus (units are ms²). Also shown is the mean local density (units are $10^{-3}$ kbp μm⁻²) (see **f**). The line indicates values for single polymer beads (each representing 1 kbp of chromatin); the shaded region indicates the s.e.m. (for the density, this is typically similar in size to the line width). Above each plot the colored block shows the input data used for each cell line as indicated: yellow are H3K27ac regions, red are binding sites inferred from ATAC-seq peaks, other regions are gray. Gene positions are indicated above the plots. Purple blocks under the genes indicate the positions of the simulated FISH probes used to obtain the MSDs in **b–d**. **f**, The local density is determined for each chromatin bead by counting the number of proteins and polymer beads within a radius of $3\sigma \approx 53$ nm.

To examine the effect of loop extrusion on dynamics, the extrusion probability ($\varphi_e$) for each bead was defined as the fraction of time in which it was in the vicinity of an extruder. In HiP-HoP, $\varphi_e$ depends only on the positions of and occupancy at CTCF-binding sites, and it shows peaks at high-occupancy CTCF sites (Extended Data Fig. 5). Extrusion does affect dynamics: there was a relatively weak negative correlation between extrusion probability and mobility (Fig. 3c). To understand how extrusion affects dynamics more generally, we performed

simulations in which we varied the extrusion rate and the density of CTCF sites (by adding new sites in random positions; Extended Data Figs. 6 and 7 and Supplementary Notes). Intriguingly, changes to mobility across the locus were subtle and difficult to predict a priori. The largest changes occurred at CTCF sites, which typically showed decreased mobility when either the extrusion rate or the number of CTCF sites was increased (Extended Data Figs. 6c and 7c). Both changes led to extruders reaching CTCF sites more quickly, with CTCF–CTCF

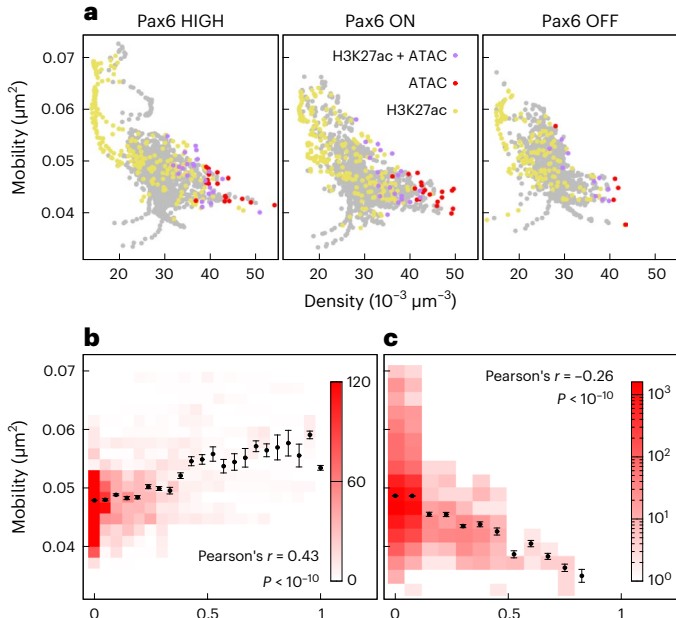

**Fig. 3 | Local density, chromatin compaction, and loop extrusion affect dynamics. a**, Scatter plots show the relationship between mobility and local density; each point represents a single polymer bead, and data from the whole 3-Mbp simulated region are included. Point color indicates the type of bead (as determined from the input data). **b**, Plot showing the relationship between mobility and H3K27ac coverage. Values for each quantity were calculated for a 20-kbp window around each 1-kbp bead (the mean value of the mobility within the window, and the fraction of the 20 kbp region covered by the H3K27ac mark are used). To obtain the color map, data were binned on both quantities; the darker the color, the more 20-kbp windows belong to the bin (color bar unit is number of windows). To obtain points, data were binned according to only the H3K27ac level, and the mean mobility was calculated for each bin. Error bars show the s.e.m. The Pearson correlation coefficient and $P$ value (two-sided Student's $t$-test, $P = 1.5 \times 10^{-13}$, based on $n = 8{,}973$ values) are shown. **c**, Plot showing the relationship between mobility and extruder occupancy. Extruder occupancy for a given bead was measured as the fraction of time in which it is bound by an extruder, and we used the mobility for each bead. Again, the color map was obtained by binning on both quantities (color bar unit is number of beads), and the points were obtained from binning on extruder occupancy with the mean mobility shown for each bin. Error bars show the s.e.m., and the Pearson correlation coefficient is indicated (two-sided Student's $t$-test, $P = 0$, based on $n = 9{,}000$ values).

loops forming more readily. Because extruder-mediated loops can stabilize (or disrupt) nearby protein-mediated loops, changing the pattern of CTCF looping leads to changes across the locus (see Supplementary Notes for further discussion).

A concern from this analysis is that the dynamics of a given chromatin bead are simply a reflection of the properties with which it is endowed in the simulation scheme. However, beads that do not overlap ATAC, H3K27ac, or CTCF peaks (that is, 'unmarked' beads) exhibited a broad range of mobility values (Extended Data Fig. 4d), indicating that the dynamics of a given bead emerge both from its own properties and those of its local environment[33].

## Dynamics show correlation with interaction locality

We hypothesized that the chromatin properties giving rise to the variation in mobility might lead to differences in chromatin interactions. A common measure extracted from 3C methods is the ratio between the amount of local and long-range chromatin interactions at a given site[34]. Although the interpretation of long-range interactions within

a 3-Mbp locus is limited, a measure of interaction 'localness' can be defined as the ratio between the number of interactions with regions within 100 kbp of the target and the number of interactions with regions farther than 100 kbp (Fig. 4a; see Supplementary Notes for details); 100 kbp was chosen because it is slightly smaller than typical promoter-enhancer loop sizes, so *cis*-regulatory loops will be counted as long-ranged (but different thresholds might be more informative if a larger locus is being considered).

Plotting localness for each 1-kbp bead across the simulated region (Fig. 4b) revealed a pronounced reduction in localness at protein-binding sites (ATAC-seq peaks, red bars), consistent with these sites being involved in bridging interactions with distant (>100 kbp) regions. Some, but not all, H3K27ac-marked regions (open chromatin, yellow shading) also displayed reduced interaction localness. There was a small, statistically significant positive correlation between localness and mobility (Fig. 4c). The trend was clear for beads with lower mobilities but did not continue for those with mobilities larger than ~0.05 μm⁻² (which had a larger spread of localness values, and possibly a non-monotonic relationship, but these constitutes <9% of the beads), consistent with protein-binding sites having low localness values, being in higher density regions, and having lower mobility.

## Locus conformations change on a timescale of minutes

The dynamics of the locus conformation as a whole can be accessed from a simulation by tracking in time the relative positions of all beads ($N$) in the chain. A more tractable approach (closer to what might be realized experimentally) is to track a smaller number of points. Here, we considered the same three FISH probe regions as above (*Pax6*, URR, and DRR) and took the three pairwise separations of these probes as a description of the locus configuration. These separations are represented by a point in a three-dimensional space (Fig. 5a): we can write a vector $\mathbf{X} = (x_{UP}, x_{UD}, x_{PD})$, where the components $x_{UP}$, $x_{UD}$, and $x_{PD}$ are the separations of the URR and *Pax6*, the URR and DRR, and the *Pax6* and DRR probes, respectively. In Figure 5b, each point represents a single instant in time in 1 of our 20 simulations of the locus in *Pax6* ON cells. The volume of the cloud of points represents the range of structures that the locus can adopt (Fig. 5d), providing a metric for 'locus variability.' The trajectory of $\mathbf{X}$ as it moves through this 'configuration space' during a single simulation can be overlaid on the scatter plot (Fig. 5c; see Extended Data Fig. 8 for examples from *Pax6* HIGH and OFF cells). Examination of these trajectories suggests that the whole volume can be explored within a single simulation of duration $7.96 \times 10^5 \tau$, equivalent to roughly 27 min—considerably shorter than a typical cell cycle (see Supplementary Notes for details). This indicates that much of the structural variability of a locus can be exhibited within a single cell.

To examine this quantitatively, a shape-change parameter ($S(t)$) was defined, which tracks the mean change in locus configuration over a lag time ($t$). Mathematically, this is the MSD of $\mathbf{X}$ as it moves through configuration space (Fig. 5e and Supplementary Notes). As expected, $S(t)$ reaches a plateau at large times, because the locus explores a finite volume within configuration space (Fig. 5f). This occurs within about 100–200 s, implying that all of the configurations are explored within this time frame. The behavior at short times, where $S$ grows as some power $\alpha$ of time (i.e., $S \propto t^\alpha$), gives information about the dynamics of the locus structure: all three cell lines show $\alpha < 1$, characteristic of sub-diffusive behavior (Fig. 5g). *Pax6* ON and HIGH have $\alpha \approx 0.5$; HIGH cells generally have a larger $S(t)$, indicating that the configuration changes more quickly. Sub-diffusion is expected, owing to the polymeric nature of the chromosome; loops being transiently stabilized by protein bridges or loop extruders likely act as dynamical traps, further inhibiting motion. The *Pax6* OFF $S(t)$ curve shows a smaller $\alpha$, perhaps reflecting the very different pattern of protein-binding sites in those cells.

How well do simulations reproduce the dynamics of real locus structures in vivo? It is likely that the dynamics depend heavily on the

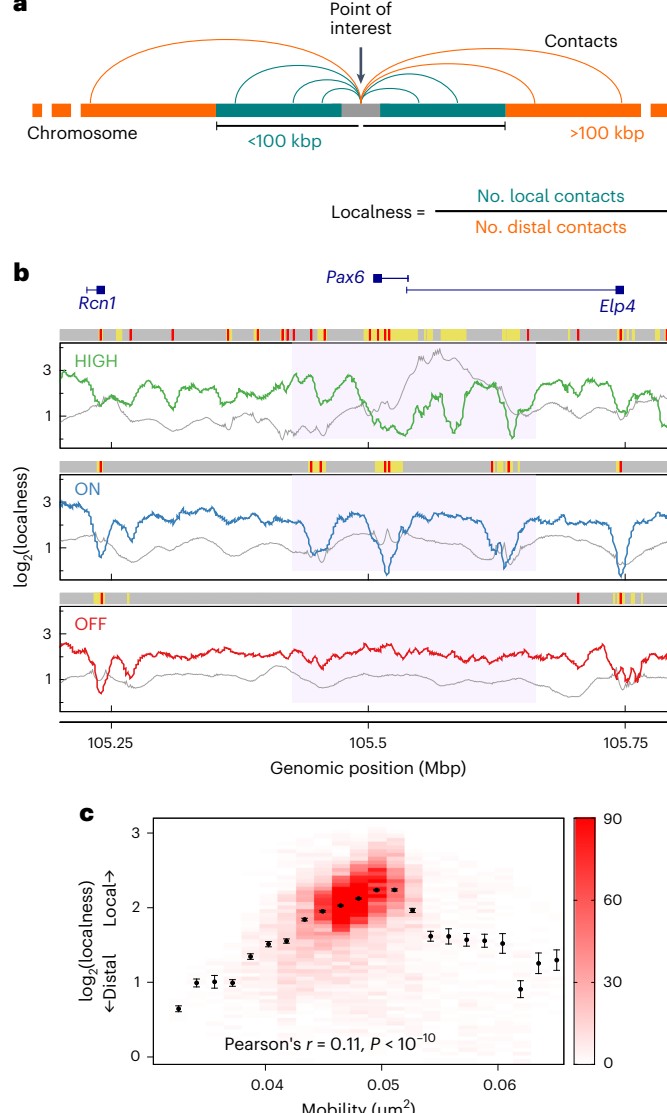

**Fig. 4 | Dynamics are correlated with interaction locality. a**, Localness is defined for a given chromatin bead as the number of interactions with regions closer than 100 kbp genomically, divided by the number of interactions with regions farther away than 100 kbp. An interaction was defined as two chromatin beads being closer together than $3.5\sigma \approx 62$ nm. **b**, Plots showing how localness of interactions varies across the locus in each cell type in simulations (colored lines; a value is shown for each 1-kbp chromatin bead). $\log_2$ values are shown. For comparison, gray lines show the mobility as in Figure 2e (a different vertical scale is used). Above each plot, the gray, yellow, and red bars indicate unmarked chromatin regions, chromatin regions marked with H3K27ac (open chromatin), or ATAC-seq peaks (binding sites), respectively, in simulations. **c**, Plot showing the relationship between mobility and localness of interactions. To obtain the color map, each chromatin bead was put into a bin on the basis of its localness and mobility; the darker the color, the more beads in the bin (color bar unit is number of beads). Points were obtained by binning each chromatin bead only according to its mobility; the mean and s.e.m. for each bin are shown. The Pearson correlation coefficient and *P* value (two-sided Student's *t*-test, $P = 8.3 \times 10^{-14}$, based on $n = 9{,}000$ values) are indicated.

model parameters, for example the number of proteins, the rate at which proteins switch between binding and non-binding states, the number of extruders, or extrusion speed[35]. So far, we have used biologically reasonable parameters that were optimized to best predict static (fixed-cell) measurements[7]. However, it is possible that there

are distinct sets of parameters that give similar static predictions, but different dynamic behavior. Although experimental determination of the parameters remains challenging, we can use the simulations to examine the effect of varying them.

First, simulations with different rates at which proteins switch between binding and non-binding states (the protein switching rate, $k_{sw}$) were performed for *Pax6* ON cells (Fig. 5h): the rate was reduced from $k_{sw} = 10^{-3}\tau^{-1} \approx 0.48$ s$^{-1}$ (gray points) to $k_{sw} = 2 \times 10^{-4}\,\tau^{-1} \approx 0.097$ s$^{-1}$ (black points). Less frequent switching led to a smaller volume in the configuration space being explored (reduced variability, Fig. 5i): configurations in which one or more of the probe pair separations is small became favored over extended configurations (reduced mean locus size). The locus also changed configurations more slowly (revealed by examining $S(t)$, Fig. 5j). These effects arise because reducing the protein switching rate leads to an increase in the size and longevity of protein clusters[26]: protein-stabilized loops are more likely to form for extended times, slowing the dynamics and favoring more compact configurations (including multi-loop 'rosette' structures).

If the switching rate is kept constant, but instead the number of proteins is decreased, there is a smaller decrease in variability (Fig. 5l), and configurations in which the probe pair separations are small are disfavored (mean locus size increases, Fig. 5k). If there are fewer proteins, there will be a lower likelihood of protein-stabilized loops. Unlike when altering $k_{sw}$, there was no change in configurational dynamics in this scenario (Fig. 5m). These results further support the idea that there is non-trivial interplay between different model ingredients. A decrease in locus variability can be accompanied by either an increase or a decrease in mean locus size and does not necessarily lead to a change in configurational dynamics. We note that the changes to the parameters led to poorer agreement with the fixed-cell experiments (CaptureC and FISH), but nevertheless provide insight into the underlying biophysical mechanisms. Interestingly, there was very little effect on $S(t)$ when loop extrusion parameters were varied (Extended Data Figs. 6d and 7d), likely owing to the specific pattern of CTCF binding around *Pax6* (loop extrusion does not seem to play a major role in promoter-enhancer interactions in this locus[7]). Other recent work using HiP-HoP suggests that loop extrusion plays an important role in cell-to-cell variability of expression[22]; it would be interesting in the future to study configurational dynamics in other loci where, for example, enhancers and promoters sit at opposite ends of a CTCF loop domain.

Finally in this section, we note that for *Pax6*, the choice to consider three probes is natural because there are two distal enhancers. One could consider more probes and examine how the system moves through a higher-dimensional configuration space. That might be more relevant for larger, more complex loci with many regulatory elements, but the general conclusions are unlikely to be affected. A complementary approach for characterizing locus dynamics is to measure time intervals between enhancer-promoter collisions and the duration of their interaction; we include such an analysis in Supplementary Notes (also Supplementary Fig. 1). A surprising result is that interaction durations are at most twofold longer in ON cells than in OFF cells; this suggests that the stabilization of loops by protein clusters is modest.

## Perturbing gene expression weakly affects locus conformation

Given the apparent links between locus configuration, gene activity, and dynamics, we performed new CaptureC and FISH experiments on *Pax6* HIGH cells after treatment with drugs to perturb either transcription or topology. Alpha amanitin was used to inhibit transcription through selective degradation of elongating polymerases[36,37]. Surprisingly, but as observed previously[38], CaptureC profiles looked very similar to those from untreated cells, with similar, if slightly higher, interaction peaks (Fig. 6a). Separations of FISH probes located at the *Pax6* promoters and two neighboring genes also showed no significant

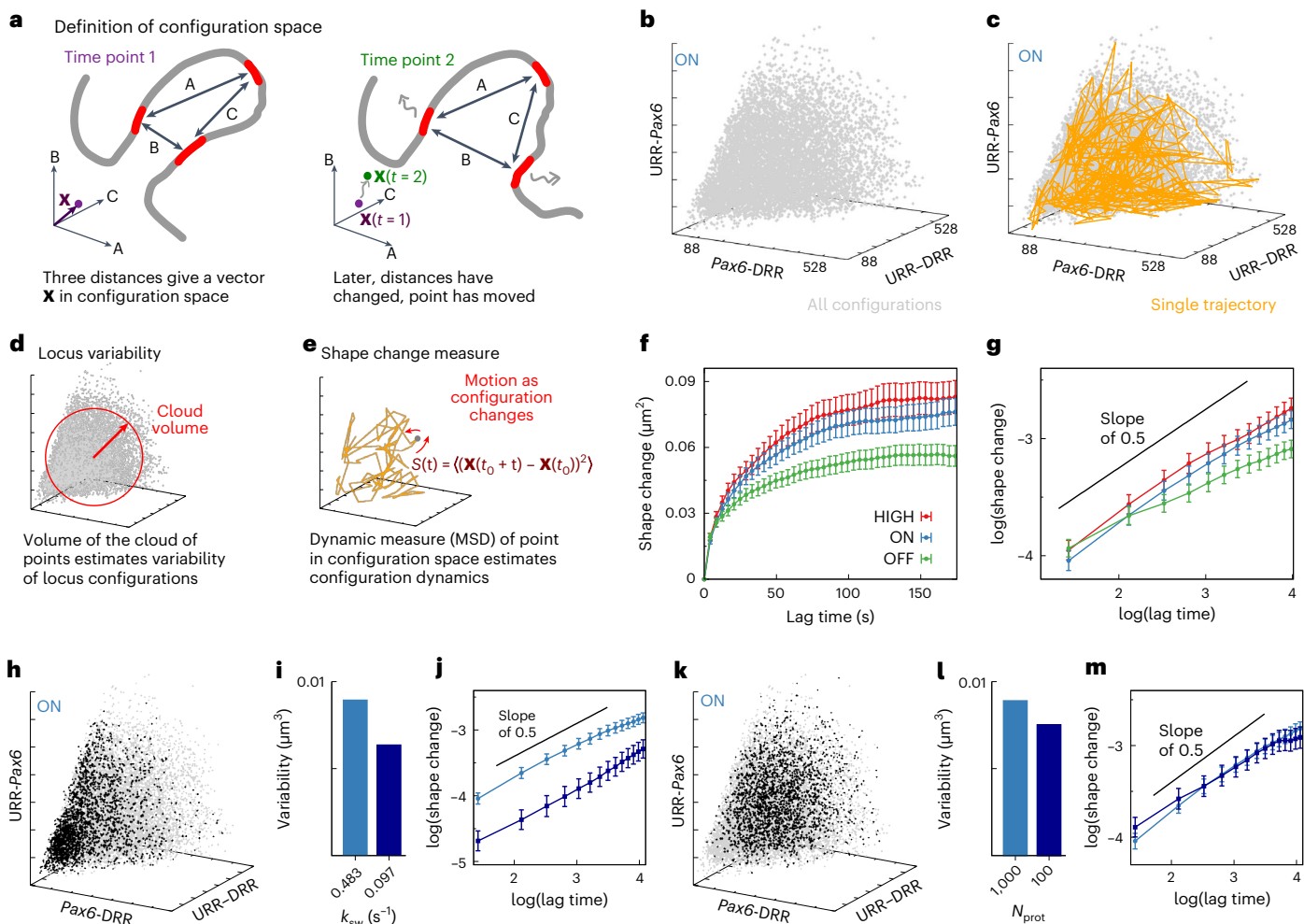

**Fig. 5 | Dynamics of locus conformation. a**, Locus conformation can be characterized using the three distances between three regions covered by FISH probes (*Pax6*, URR, and DRR). These can be considered as a point in a three-dimensional configuration space. As the configuration changes in time, this point moves in configuration space. **b**, Scatter plot showing configurations adopted in *Pax6* ON cells. Each point represents a single time point; data are shown for 20 repetitions of the simulation. Units are nm, and all axes show the same range. **c**, The same scatter plot as in **b**, but a trajectory from a single representative simulation is overlaid. The time interval between points along the path is equivalent to ~4 s; the total simulation duration is ~27 min. **d**, The size of the cloud of points represents the variability of the locus. **e**, The trajectory of a point through configuration space (**X**(t)) can be used to quantify how quickly locus conformation changes via a shape-change parameter. **f**, The shape-change parameter as a function of lag time. Error bars show s.e.m. (for each point, the

number of independent measurements is at least $n = 3,570$). **g**, The same data as in **f** are shown on a logarithmic scale (error bars show s.e.m.). The black line has a slope of 0.5. **h**, Scatter plot of configurations for *Pax6* ON from simulations with a reduced protein switching rate (black points show $k_{sw} \approx 0.097$ s$^{-1}$; gray points show $k_{sw} \approx 0.49$ s$^{-1}$, as in **b**). Axis ranges are the same as in b. **i**, Bar plot showing how locus variability changes with switching rate. **j**, The shape-change parameter is shown for *Pax6* ON from simulations with different switching rates. Error bars show s.e.m. (the number of independent measurements is at least $n = 1,850$). **k**, Similar plot to that in **h**, but showing simulations with 100 proteins (black) and 1,000 proteins as in **a** (gray); $k_{sw} \approx 0.49$ s$^{-1}$ in both cases. **l**, Bar plot showing how the number of proteins affects locus variability ($k_{sw} \approx 0.49$ s$^{-1}$). **m**, The shape-change parameter is shown for *Pax6* ON from simulations with different numbers of proteins. Error bars show the s.e.m. (the number of independent measurements is at least $n = 1,850$).

change after treatment (Fig. 6b). Together, these findings suggest that, at least at *Pax6*, inhibiting transcription per se does not greatly affect structure, but frequently occurring interactions (CaptureC peaks) are enhanced.

To interpret these results, we considered how transcription inhibition could be modeled within the simulations. A loss of transcription could be implemented as a loss of protein binding, because our model proteins represent complexes of polymerase and transcription factors (Fig. 6c). However, a simulation in which all binding sites were switched off showed a dramatic loss of many interaction peaks in the simulated CaptureC, and a general decrease in interactions, an effect opposite to that observed experimentally (Fig. 6d).

In HiP-HoP, the bridging complexes continually switch between a binding and a non-binding state[26] to model chemical reactions

(for example, post-translational modifications) and provide a realistic turnover of proteins in foci. Another possible effect of alpha amanitin is that it might interrupt such reactions, for example, in the polymerase transcription cycle, and polymerases could become stuck in an initiation state. A modified simulation was used to test this: all protein-chromatin bonds were 'fixed,' so that from the point of simulated alpha amanitin treatment, any protein bound to a chromatin bead remained bound (Fig. 6e and Supplementary Notes). This led to an increase in interactions at many peaks in the simulated CaptureC (Fig. 6f), a situation closer to the experimental observations than in the scenario in which proteins were removed.

Simulated FISH measurements showed different trends for each of the two models (Fig. 6g). Removing proteins led to increased probe pair separations, whereas fixing protein-chromatin bonds led to a decrease.

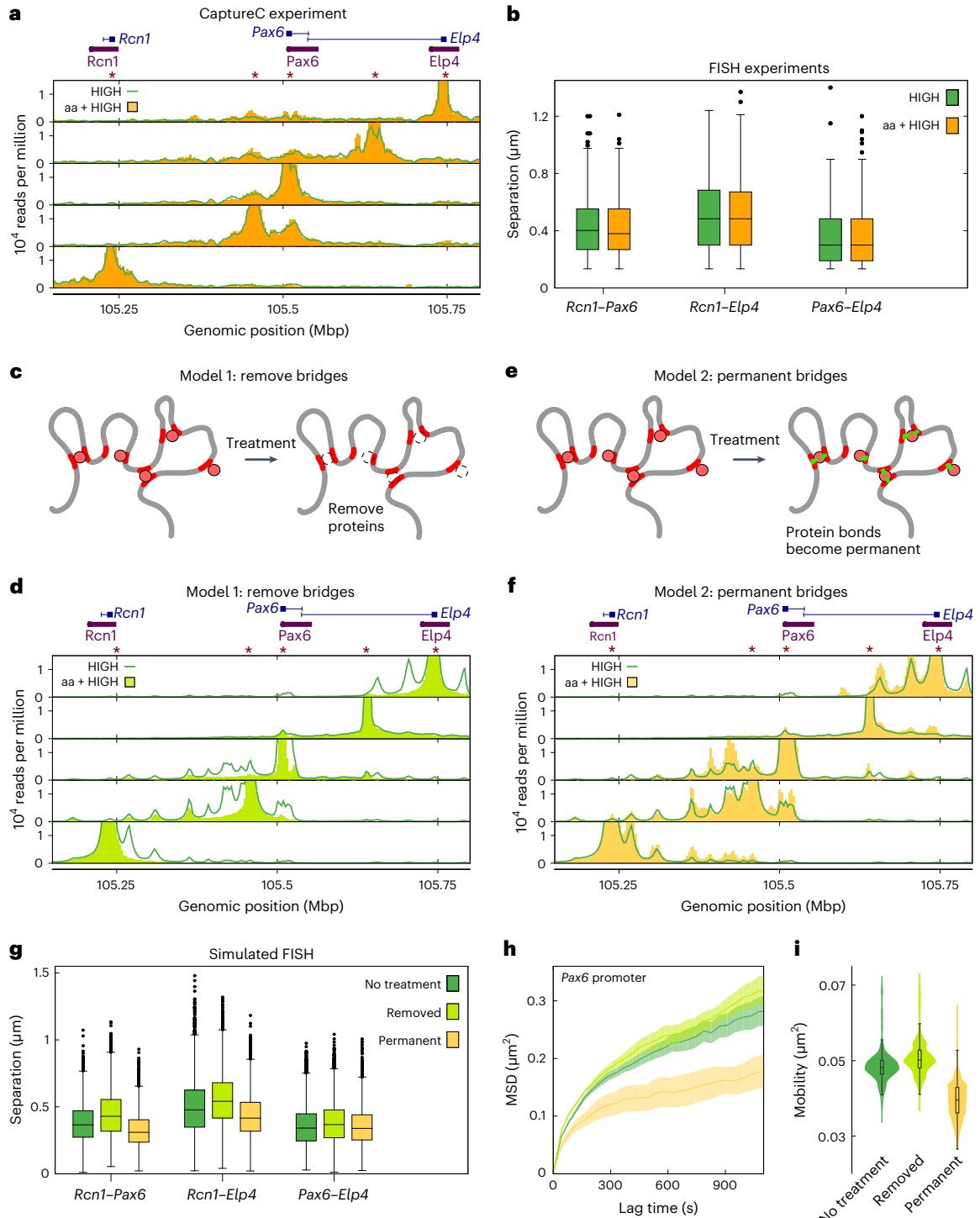

**Fig. 6 | Perturbing transcription weakly affects locus conformation.**
**a**, CaptureC data from *Pax6* HIGH cells in an experiment involving alpha
amanitin (aa) treatment are shown (yellow bars) alongside data from untreated
cells (green line). **b**, FISH data for alpha-amanitin-treated and untreated *Pax6*
HIGH cells. Probes were positioned over *Pax6* and the promoters of the two
adjacent genes *Rcn1* and *Elp4* (purple bars in **a** and probe set 2 in Fig. 1b).
Box plots show the distributions of each separation (from left to right, the number
of measurements in each case are $n$ = 147, 142, 125, 131, 130, 131). Here and in
later panels, box limits give the interquartile range with whiskers extending
by a factor of 1.5 and the center line giving the median. A Mann–Whitney $U$ test
did not reject the null hypothesis that treated and non-treated distances were
drawn from the same distribution ($P$ > 0.05 for all three pairs of probes; from
left to right, $P$ = 0.87, $P$ = 0.44, and $P$ = 0.53). **c**, In one possible model of alpha
amanitin treatment, all proteins are removed, because these could represent

complexes containing RNA polymerase (see text). **d**, Simulated CaptureC for
*Pax6* HIGH cells from simulations in which all protein-binding sites were removed
(lime green bars). Equivalent results without the simulated treatment are also
shown (green line). **e**, In an alternative model, at the point of alpha amanitin
treatment, all protein-chromatin bonds are fixed (see text and Supplementary
Notes). **f**, Simulated CaptureC for *Pax6* HIGH cells from simulations in which
all protein-chromatin bonds were fixed at the point of treatment (yellow bars).
**g**, Plot showing simulated FISH measurements for the two models of alpha
amanitin treatment, as well as the untreated case (the number of measurements
in each case is $n$ = 7,980). **h**, The MSD of the *Pax6* promoter FISH probe is shown
for the three cases; shaded bands show the s.e.m. **i**, The mobility is calculated for
each chromatin bead within the locus. The distributions of mobility values are
shown as violin plots, with overlaid boxes showing the median and interquartile
range, with whiskers extending by a factor of 1.5.

Locus dynamics also changed for both scenarios (Fig. 6h,i): protein removal led to faster dynamics, whereas protein fixing led to slowed dynamics. Recent particle-tracking experiments in which histone proteins were labeled fluorescently have indicated that chromatin is typically more dynamic after alpha amanitin treatment[39].

Since neither model fully predicts both FISH and CaptureC data, we suggest that alpha amanitin treatment actually results in a mixture of these effects: loss of some bridging (for example, mediated by complexes involving elongating polymerase) results in increased mobility and local decompaction, but some other enhancer-promoter bridging (for example, involving paused polymerases) may be stabilized, resulting in more pronounced CaptureC peaks. A proper representation of the action of alpha amanitin in simulations would require a more explicit inclusion of transcription than is currently possible within HiP-HoP.

In a complementary experiment, *Pax6* HIGH cells were treated with bleomycin to perturb the local chromatin topology (DNA is nicked, releasing superhelical tension so that it becomes torsionally relaxed[37]). After perturbation, CaptureC profiles looked very similar to those from untreated cells, whereas FISH revealed no significant change (Extended Data Fig. 9). This suggests that at this scale of analysis, superhelical tension, which has dramatic effects on larger scale chromatin structure[37], does not affect chromatin architecture within the locus.

## Discussion

In this work, we extended the HiP-HoP chromatin modeling scheme to study gene locus dynamics. Chromatin mobility can be determined by extracting MSDs from simulations. The observed dynamics depend strongly on the size of the chromatin segment being tracked. Chromatin neighborhoods (35–40-kbp regions) at different points within a locus, or in different cell lines, showed highly similar dynamics; by contrast, different 1-kbp regions could have very different mobilities. This was dependent on two factors: local chromatin fiber disruption (marked by H3K27ac), and bridging interactions between regulatory elements. Typically, disrupted chromatin[20] was more dynamic than compact chromatin, whereas regions enriched in protein-mediated loops, often marked by ATAC-seq peaks, were less dynamic. However, active binding sites tend to be embedded within H3K27ac regions, leading to a balance between the bridging-mediated slowing and the disruption-mediated speeding up of motion; the dynamics of a segment depend on the surrounding environment as well as its own properties.

The prediction that the position and size of a probe determine the observed dynamics will be critical when designing future live-cell imaging experiments, and could explain some previous contradictory results. In live-cell imaging in which single enhancers and promoters were labeled using CRISPR–dCas9 with guide RNAs spanning ~2 kbp, these became more mobile when active[40]. Conversely, experiments using an ANCHOR/ParB DNA-labeling system to track a promoter showed motion becoming constrained upon activation[41]. Several studies tracking multiple points across the genome (for example, by labeling histones) showed that dynamics tend to increase as activity is reduced (by transcription inhibition, serum starvation, or RNA polymerase II (Pol II) degradation[8,32,39]). These observations can be reconciled if, as our model suggests, gene activation leads to chromatin decompaction (increasing mobility) in some regions, but additional looping (decreasing mobility) in others.

The concept of promoter-enhancer interactions often leads to a notion of somewhat static gene locus configurations; by contrast, our study introduces the concept of sampling between regulatory elements, where interactions continually change over time. Our results indicate that, for a typical locus, many possible interaction events can be sampled within tens of minutes (Fig. 5), suggesting that although promoter-enhancer contacts revealed by 3C are enriched

above a background, there is continuous sampling. Importantly, we found that, within a simulation equivalent to ~30 min, the locus was able to explore all of its configuration space. Dynamics depend on the model parameters, but our chosen parameter set is validated by comparison to fixed-cell experiments. Varying the number of active proteins and the rate at which these switched between binding and non-binding states[26] had subtle effects. Intriguingly, we found that different changes to the parameters that have the same effect on static properties do not necessarily have the same effect on dynamics. We note that configurational dynamics will also depend strongly on locus size: we would expect, for example, a 1-Mbp TAD to take longer to re-organize than the ~200-kbp region studied here. This is consistent with previous modeling based on fitting to 5C data[42] that suggested that multiple TAD configuration states should be dynamically accessible during a cell cycle.

We found that inhibiting transcription using alpha amanitin did not greatly alter *Pax6* locus structure, leading only to small increases in CaptureC interaction peak heights. Our experiments stand in contrast with recent Hi-C, HiChIP, and OCEAN-C data showing that acute degradation of Pol II led to a small decrease in looping interactions[43]; another study using MicroC in mice found that transcription inhibition using triptolide or flavopiridol did not greatly affect promoter-enhancer interactions, but other features in the data were affected[44]. Other work showed that alpha amanitin did not disrupt enhancer interaction hubs[38], and that elongation inhibition via DRD treatment did not disrupt Pol II foci[45]. Together, these results suggest that there is a complicated relationship between transcription and chromatin contacts and point to subtleties in the action of different inhibitors, which are not understood.

These simulations of the *Pax6* locus give insight into the factors affecting chromatin dynamics and suggest that for a locus of this size, in a given time, a single cell would show the same level of variability as that observed from cell to cell across a population. We expect that a similar order of magnitude for the timescale for locus rearrangement would be found in other polymer models for chromatin (for example, in simpler models with fewer 'mechanistic ingredients'[11,14,16]); however, our results suggest that differences across a locus or between loci depend on details of the model (that is, on which local chromatin features are included). In the future, it would be interesting to simulate larger regions or to include additional model ingredients that might shed light on other observations made in live cells (for example, correlated chromatin motion[8,32], dependence on nuclear structural proteins like lamins[46], and gel-like features of chromatin[47–49]). HiP-HoP could be extended to include, for example, repressive proteins that compact DNA, nuclear lamina interactions, or proteins such as SAF-A that are thought to form an RNA-dependent gel constraining chromatin motion[47].

## Online content

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

## Methods

### HiP-HoP simulations

The HiP-HoP model was used as previously described[7]. In brief, a chain of beads connected by springs represented a chromosome region, with 1 kbp of DNA per bead. Diffusing spheres represent complexes of proteins which bind at active chromatin sites, and these switch back and forward between a binding and a non-binding state with rate $k_{sw} = 10^{-3}\,\tau^{-1}$ (unless otherwise stated), where $\tau$ is the simulation time unit. DNA accessibility data (ATAC-seq) were used to identify binding sites. Loop extrusion was modeled by introducing additional springs between adjacent beads in the chain, which were then moved at regular time intervals to extrude a loop. Extruders were initiated at random positions, moved at a rate $k_{ex} = 2\,\text{bp}\,\tau^{-1}$, and were removed with rate $k_{off} = 2.5 \times 10^{-5}\,\tau^{-1}$. ChIP followed by microarray hybridization (ChIP–chip) data for CTCF and Rad21 binding were used to identify loop anchor sites, and extrusion was halted at loop anchor sites in a direction-dependent manner. In some regions of the bead chain, additional springs were added to 'crumple' it into a more compact structure. Histone modification data (H3K27ac ChIP–chip) were used to identify regions that do not have the crumpling springs. The dynamics were evolved through a Langevin dynamics scheme (implicit solvent) using the LAMMPS molecular dynamics software[50]. In our previous work[7], the simulations matched the average diffusion properties of the model chromatin to experimental measurements, but in that case only a short region of the genome was simulated, so spatial variations in chromatin density were not accurately represented. In this work, a much larger chromatin fragment (40 Mbp) was simulated to better match the overall density found in vivo. For efficiency, 10 instances of a 3-Mbp region around *Pax6* (chr2:104,000,000–107,000,000 mm9 reference genome) were placed along this 40-Mbp (40,000 bead) fragment. Periodic boundary conditions were used, and the system size was chosen to give a roughly realistic chromatin density. We used data from 3 mouse cell lines, and in a single simulation, a mixture of loci from the 3 were included among the 10 instances. In repeat simulations, the positions of the loci in different cell lines were randomized (meaning that, on average, each copy of the locus was embedded in a similar chromatin context). As noted above, to ensure that this new set up still gave good predictions of locus conformation, results were compared with both CaptureC data and FISH (Extended Data Figs. 1 and 2).

In the simulations, time is evolved in discrete steps, with a simulation 'time unit' $\tau$, which can be mapped to a real time by comparison with experiments. Typically, each individual simulation was run through 1.5 million simulation time units. We initialized the fiber in a configuration resembling a mitotic chromosome; during the first part of the simulation, the system relaxes to a realistic configuration representative of the in vivo locus in interphase. We therefore discarded the first $7 \times 10^5\,\tau$ of each simulation, after which time the dynamics had reached a steady state (see Supplementary Notes). The locus configuration varied dynamically through the remaining 800,000 $\tau$, and we determined that extracting 400 configurations at regular time intervals provided a good representation of the locus motion over this time (see Supplementary Notes). We generated 20 trajectories (representing single cells) for each cell line, extracting a total of 8,000 configurations per cell line. Full details of the simulations, interaction potentials, and all parameters are given in the Supplementary Notes. The input data were previously published[7] and are available at GEO: GSE119660, GSE119656, GSE119659, GSE119658, GSE120665, and GSE120666. Full details of the data analysis are given in Supplementary Notes.

### Mapping simulation length and time scales to real units

Simulation length and time units were mapped to physical units by comparison with experimental data. To estimate the length unit, for a given pair of FISH probes, a distribution of separations was obtained from simulations and experiments[7], and compared using the Kolmogorov–Smirnov statistic (the smaller this statistic the closer the two distributions). We obtained nine such distributions (three distances between *Pax6*, URR, and DRR probes in each of three cell lines), and used the mapping that minimized the average of the nine Kolmogorov–Smirnov statistics. This gave an estimate for the length unit of $\sigma \approx 17.6\,\text{nm}$. To map the simulation time unit, we calculated an MSD for every chromatin bead in every simulation, and obtained an average. This was compared with data from motion-tracking experiments in ref. 30, where MSDs were obtained for several chromosome regions in *Saccharomyces cerevisiae*. We used a linear fit to find the mapping that minimized the difference between the simulated and experimental MSD curves. This led to an estimate for the simulation time unit of $\tau \approx 2.07 \times 10^{-3}\,\text{s}$.

### Cell culture

*Pax6* HIGH cells (also known as β-TC3 cells, obtained from DSMZ, cat. no. ACC-324, RRID: CVCL_0172) were isolated from a mouse insulinoma[51] and were cultured in Dulbecco's Modified Eagle Medium (DMEM) (Thermo Fisher) supplemented with 10% fetal calf serum and 1% penicillin–streptomycin at 37 °C in 5% $CO_2$. No cell authentication was performed, and the sex of the cell line is not known.

### Alpha amanitin and bleomycin treatment experiments

To block transcription, *Pax6* HIGH cells were plated in a 6-cm dish and, once at ~80–90% confluence, treated with 100 μM alpha amanitin (Sigma A2263-1MG) for 7 h or with mock treatment (PBS alone). Inhibition was assessed by real-time quantitative reverse transcription PCR (qRT–PCR); samples were washed with PBS, and RNA was extracted using RNeasy mini-Kit (Qiagen). All RNA samples were treated with on-column DNase1 treatment (Qiagen); the RNA concentration was corrected across the sample and reverse transcribed to complementary DNA (cDNA) using SuperScript II (Thermo Fisher) standard first-strand synthesis protocol with oligo(dT) primers (Promega), with two biological replicates per condition. Real-time qPCR on cDNA was performed using a LightCycler 480 II (Roche) and SYBR Select Master Mix (Thermo Fisher), using the standard manufacturer's protocol. Gene-specific primers were designed for intron-exon junctions to assay nascent RNA, and the fold change against mock treatment control was calculated using the $2^{-\Delta\Delta CT}$ method, normalizing against 18S as a housekeeping gene.

For bleomycin treatment, *Pax6* HIGH cells at 80–90% confluency were trypsinized to a single cell suspension; $1 \times 10^6$ cells were treated with 250 μM of bleomycin (Cayman Chemical) for 15 min in PBS, centrifuged to remove the bleomycin, and washed with PBS. Samples were processed using a standard protocol for genomic DNA preparation from a Pure link genomic DNA extraction kit (Invitrogen). To assay DNA nicking, a comparison between 250 μM bleomycin and PBS treatment alone was performed on 1 μg extracted genomic DNA and run on an alkaline agarose gel under denaturing conditions, before staining with ethidium bromide and imaging, using a standard alkaline gel protocol[52].

### CaptureC experiments

For *Pax6* HIGH cells that had been treated with alpha amanitin or bleomycin, NG CaptureC was performed as previously described[28,29], but with the following alterations. Two replicates of $5 \times 10^6$ cells were processed for each case; cells were fixed with 2% formaldehyde and lysed with standard 3C lysis buffer for 15 min before being snap frozen. Cells were further lysed by re-suspension in water and then in 0.5% SDS for 10 min at 62 °C. Each replicate was split between three tubes, re-suspended in 800 μL 1× *DpnII* buffer (NEB) with 1.6% Triton X-100, and digested with 3 sequential additions of 750 units *DpnII* enzyme at 37 °C with 1,200 r.p.m. shaking over 24 h. Samples were heat inactivated at 65 °C for 20 min, and 3 samples from each replicate were combined into 7 mL with 1× T4 DNA Ligase Buffer (NEB), with 1% Triton X-100 and

                                    

12,000 units of T4 DNA ligase at 16 °C overnight. Samples were treated with Proteinase K overnight at 65 °C and RNase A/T1 (Thermo Fisher) for 1 hour at 37 °C, before a standard phenol–chloroform extraction and ethanol precipitation was performed. Complete digestion and ligation were assessed by gel electrophoresis.

Purified 3C DNA from each sample was sonicated to 200–400 bp with a Soniprep 150 probe sonicator at 4 °C and purified with a standard Ampure XP Bead protocol (Beckman Coulter) using a 1/1.5 DNA to bead ratio. Two Illumina sequencing libraries were prepared per capture pool replicate, with 6 µg of starting DNA in each, and generated using NEBNext DNA Library Prep Kit (NEB). Samples were indexed with unique barcodes using NEBNext Multiplex Oligos for Illumina (NEB). Two separate capture pools were designed to the following *Pax6* locus elements, as in ref. 7 (a list of targeted restriction enzyme fragments is given in Supplementary Table 1). Capture oligonucleotides were designed for each end of the targeted *DpnII* fragments[29], and each was synthesized in a separate 4 nM synthesis, with a 5′ biotin label on a 120-bp Ultramer (IDT). Capture oligonucleotides from each of the two pools were mixed at equimolar amounts and pooled to a final concentration of 13 pmol in a volume of 4.5 µL per sequence capture. Libraries were sized and quality controlled on a D1000 Tapestation tape (Agilent).

NG CaptureC sequence capture was performed using SeqCap EZ HE-Oligo Kit A or B (dependent on the multiplex barcode) and SeqCap EZ Accessory Kit (Nimblegen)[29], using each of the two capture pools, with 1.5–2 µg 3C library DNA per hybridization reaction. Each hybridization reaction was performed on a thermocycler at 47 °C and incubated for between 66 and 72 h. Each hybridization reaction was then bound to streptavidin beads from SeqCap EZ Pure Capture Bead Kit and washed with SeqCap EZ Hybridization and Wash Kit (Nimblegen), following the manufacturer's protocol. Hybridization reactions were split into two and libraries were re-amplified using Post LM-PCR oligonucleotides (Nimblegen) and Q5 High-Fidelity DNA polymerase (NEB) directly from the beads, and then the DNA was purified using Ampure XP Beads, with a 1/1.8 DNA to bead ratio. A second hybridization reaction was performed as above on the re-amplified 3C libraries with 2 reactions pooled together (~1 µg in each) and incubated for 22–24 h. Washed and re-amplified double-captured libraries were sized and quality-controlled on a D1000 Tapestation tape (Agilent), and paired-end sequenced on an Illumina Hi-seq 2500 or Hi-seq 4000.

CaptureC data were analyzed using the capC-MAP software[53] (see Supplementary Notes for further details).

### Three-dimensional DNA fluorescence in situ hybridization
Cells were grown overnight on glass slides. Slides were rinsed with PBS and fixed in 4% paraformaldehyde for 10 min. Slides were rinsed with PBS and cells were permeabilized for 10 min on ice with PBS supplemented with 0.2% Triton X-100. After rinsing, slides were stored in 70% ethanol at 4 °C.

For processing, slides were dehydrated through an ethanol series and incubated with 2× SSC supplemented with 100 µg ml⁻¹ RNase A (Invitrogen) at 37 °C for 60 min. Slides were then rinsed briefly with 2× SSC, dehydrated through an ethanol series, and air dried. Slides were warmed by incubation in a 70 °C oven for 5 min before denaturation for 1 min in 70% formamide in 2× SSC, pH 7.5, at 70 °C. Slides were then transferred to 70% ethanol on ice, dehydrated through an ethanol series, and air dried before overnight hybridization at 37 °C with pairs of fosmid probes (listed in Supplementary Table 2). Probes (BacPac resources) were labeled in green-500-dUTP (ENZO life sciences), digoxigenin-11-UTP (Roche), or biotin-16-dUTP (Roche). Then, 150 ng of each labeled probe was hybridized with 5 µg salmon sperm and 10 µg human Cot1 DNA. Slides were washed 4 times for 3 min in 2× SSC at 45 °C and 4 times for 3 min in 0.1× SSC at 60 °C before being transferred to 4× SSC with 0.1% Tween 20 at room temperature. Digoxigenin-labeled probes were detected using one layer of rhodamine-conjugated

sheep anti-digoxigenin and a second layer of Texas-red-conjugated anti-sheep (Vector Laboratories). Biotin-labeled probes were detected using one layer of FITC-conjugated streptavidin followed by a layer of biotin-conjugated anti-avidin and a second layer of FITC-conjugated streptavidin (Vector Laboratories). Slides were counter-stained with 0.5 µg ml⁻¹ DAPI.

### Image capture and analysis
Three-color 3D DNA FISH slides were imaged using a Hamamatsu Orca AG CCD camera (Hamamatsu Photonics) Zeiss Axioplan II fluorescence microscope with Plan-Neofluar objectives, a 100-W Hg source (Carl Zeiss), and a Chroma 83000 triple band-pass filter with single excitation filters installed in motorized filter wheels (Prior Scientific Instruments). Image capture and analysis were done using in-house scripts written for Iola Spectrum (Scanalytics). For FISH, images were collected from at least 50 randomly selected nuclei for each experiment and then analyzed using custom Iola scripts that calculate the distance between two probe signals.

### Reporting summary
Further information on research design is available in the Nature Portfolio Reporting Summary linked to this article.

### Data availability
This work makes use of previously published publicly available data sets which are available via NCBI's Gene Expression Omnibus through the following GEO Series accession numbers. ATAC-seq data are available at GEO GSE119656; ChIP–chip data for H3K27ac, CTCF, and Rad21 are available at GEO GSE119659, GSE119658, and GSE120665, respectively; CaptureC data available at GEO GSE120666. New sequencing data (CaptureC) generated for this work are available at GEO GSE235334 (alpha amanitin treatment with control) and GSE235335 (bleomycin treatment with control). All data underlying the figures, together with the FISH and simulation data are available[54] via the Edinburgh DataShare repository https://doi.org/10.7488/ds/7477. Sequencing data were aligned to the mouse mm9 build reference genome, which was obtained from the UCSC Genome Browser website (https://hgdownload.soe.ucsc.edu/goldenPath/mm9/bigZips/mm9.2bit).

### Code availability
This work uses the LAMMPS molecular dynamics software package (www.lammps.org), along with some custom scripts developed as part of the HiP-HoP model[7]; previously published scripts are available[55] via the Edinburgh DataShare repository https://doi.org/10.7488/ds/2434. New scripts and analysis code generated in this project are available[54] via the Edinburgh DataShare repository https://doi.org/10.7488/ds/7477.

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

## Acknowledgements

We acknowledge support from European Research Council, grant number CoG 648050 THREEDCELLPHYSICS (D.M.), and the UKRI Medical Research Council, grant number MC_UU_00007/13 (N.G.). We also acknowledge M. Chiang and C. Battaglia for useful discussions. This research was funded in whole, or in part, by the European Research Council (CoG 648050, THREEDCELLPHYSICS) and the UK Medical Research Council (MC_UU_00007/13). For open access, the author has applied a creative commons attribution (CC BY) licence to any author accepted manuscript version arising.

## Author contributions

G.F. performed the simulations, and G.F. and C.A.B. analyzed the data. A.B. performed the experiments, and A.B. and S.B. analyzed the microscopy data. D.M., N.G., and C.A.B. designed the research, and all authors contributed to writing the manuscript.

## Competing interests

The authors declare no competing interests.

## Additional information

**Extended data** is available for this paper at https://doi.org/10.1038/s41594-023-01059-8.

**Correspondence and requests for materials** should be addressed to Nick Gilbert or Chris A. Brackley.

**Peer review information** *Nature Structural & Molecular Biology* thanks the anonymous reviewers for their contribution to the peer review of this work. Beth Moorefield, Carolina Perdigoto, and Dimitris Typas were the primary editors on this article and managed its editorial process and peer review in collaboration with the rest of the editorial team. Peer reviewer reports are available.

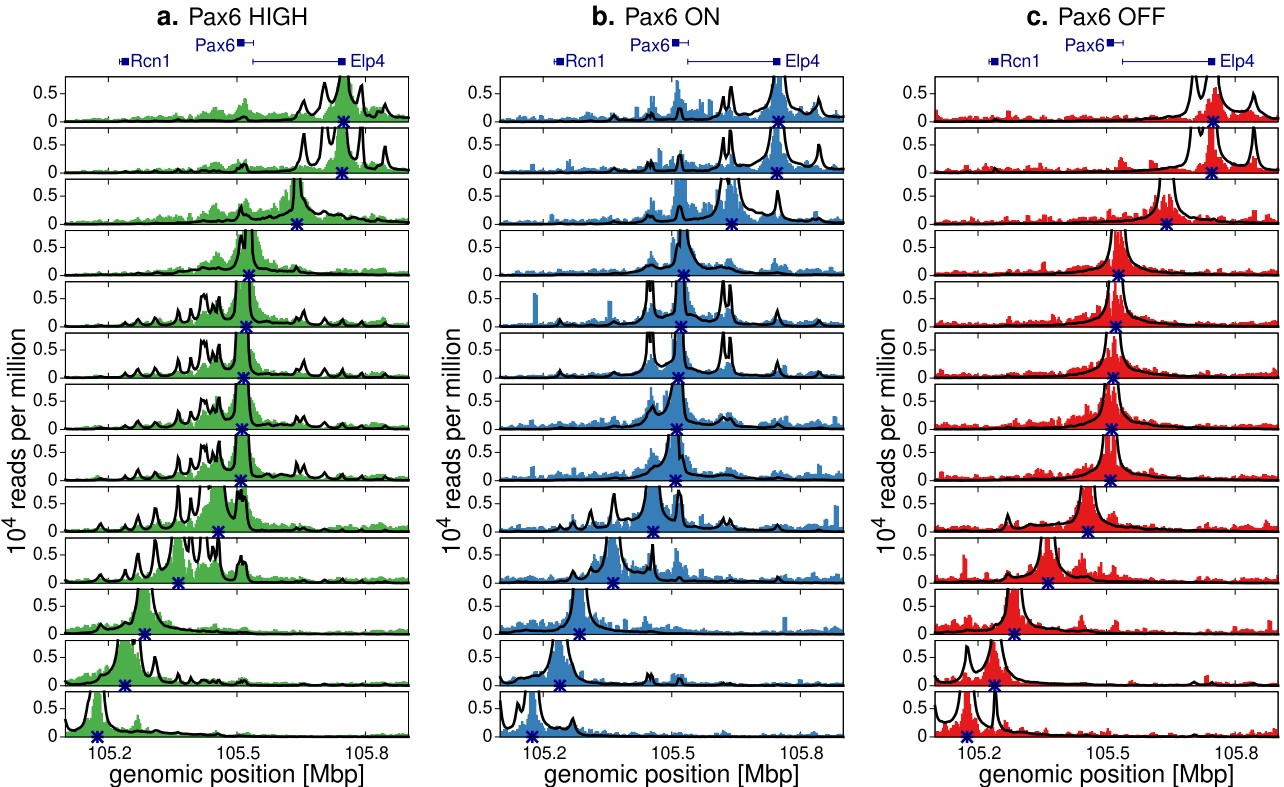

**Extended Data Fig. 1 | Simulations reproduce CaptureC data from viewpoints across the locus.** Experimental data[12] (shaded regions) are shown on the same axes as simulation results. Dark blue asterisks on the horizontal axes mark the positions of the viewpoints. Positions of the genes are indicated above the plots.

Results are shown for: **a**, *Pax6* HIGH cells; **b**, *Pax6* ON cells; and **c**, *Pax6* OFF cells. A similar level of agreement between simulations was observed as in Ref. 12, where single copies of the locus were simulated in isolation.

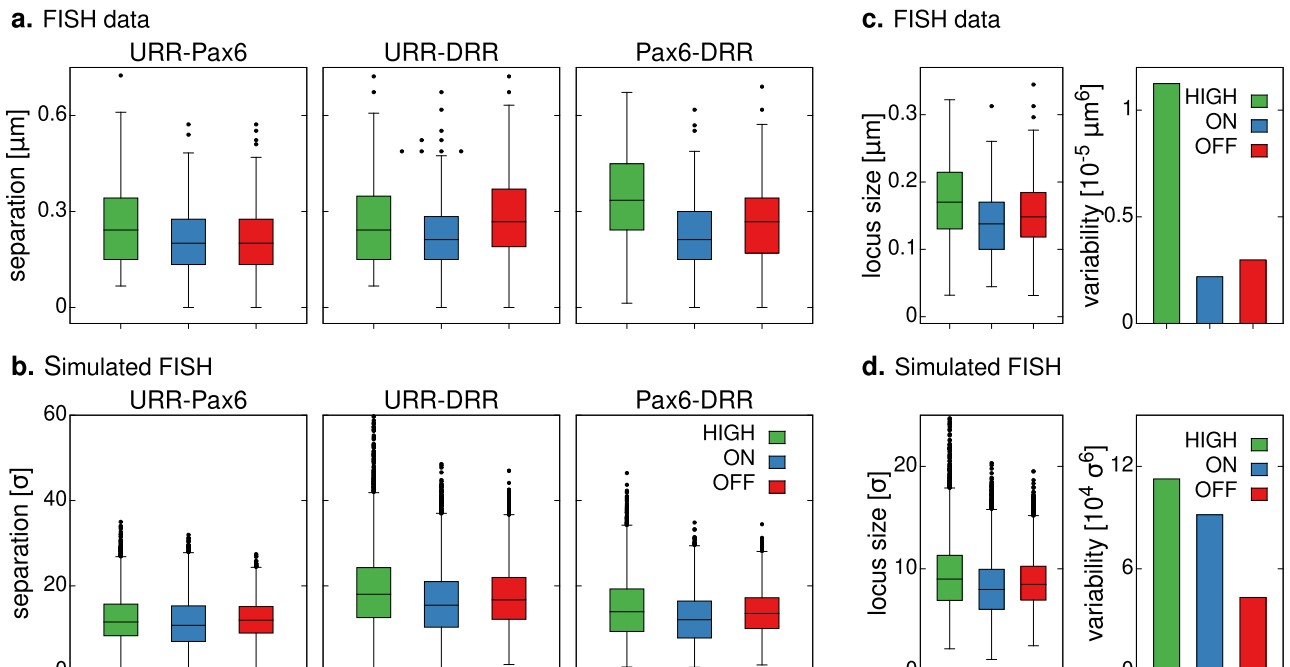

**Extended Data Fig. 2 | Simulations reproduce trends observed via microscopy. a**, FISH microscopy data was obtained from Ref. 12. Box plots show separation measurements from three pairs of FISH probes, positioned at *Pax6*, the upstream regulatory region (URR), and downstream regulatory region (URR). The number of measurements for all pairs of probes was $n = 113$, $n = 140$, and $n = 153$ for *Pax6* HIGH, ON and OFF respectively. Here and in later panels, box extents give the interquartile range with whiskers extending by a factor of 1.5 and the centre line giving the median. **b**, Equivalent measurements obtained from simulations ($n = 7980$ independent measurements in all cases). Similar trends were observed as in experiments. Specifically, *Pax6* HIGH cells were the least compact (largest separations on average), while *Pax6* ON cells were the most compact (smallest separations on average). A similar level of agreement was observed as in Ref. 12. Simulation length units ($\sigma$) are used; these can be

mapped to real lengths by comparison with the data in panel a, as detailed in Supplementary Notes. **c**, Three-color FISH microscopy allowed separations between all three probes to be measured in a given cell. This enabled the overall locus size to be determined (as defined in Supplementary Notes; the number of independent measurements for each cell line is as quoted for panel a). A measure of the cell-to-cell variability was also defined (see Supplementary Notes). **d**, Similar measures from the simulations show that the trends for locus size are captured ($n = 7980$ independent locus size measurements in each case). The trend for the variability is not correctly captured in the simulations, and this is a less good agreement than in Ref. 12. This suggests that the variability is also affected by the broader surrounding environment of the locus, which is different here than in our previous work (that is, each simulation includes multiple copies of the locus).

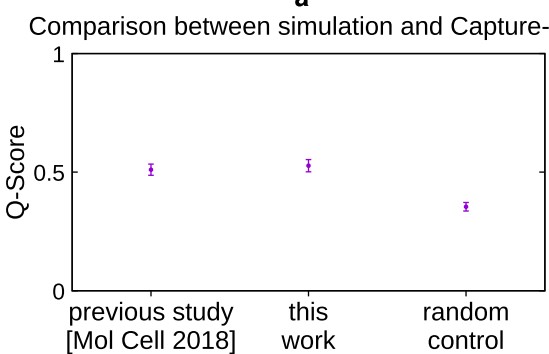

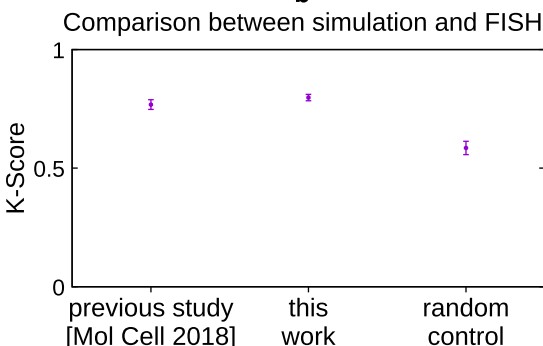

**Extended Data Fig. 3 | Quantitative comparison between simulations and experiments.** As in Ref. 12, we defined two comparison metrics, the Q-score and K-score, which quantify the level of agreement between simulations and Capture-C and FISH data respectively (see Supplementary Notes for details). **a**, Plot showing the Q-score, which compares simulated Capture-C with experiments. Points give the Q-score averaged over different Capture-C profiles (13 viewpoints in each of the three cell lines). Values range between 0 and 1, with a value of 1 indicating perfect overlap between simulation and experimental interaction peaks. Error bars show the standard error in the mean (s.e.m., $n = 39$ measurements in each case). We compare agreement between new simulations performed in the present work, and previous simulations from Ref. 12. For context, a value for a random control is also shown. **b**, Similar plot showing the K-score, which compares simulated FISH distributions with experiments. This is based on the Kolmogorov-Smirnov statistic, and takes values between 0 and 1, where 1 indicates a complete overlap of all simulated and experimental distributions. Points represent an average over scores for each distribution of probe pair separations in each cell type; error bars show s.e.m. ($n = 18$ measurements in each case).

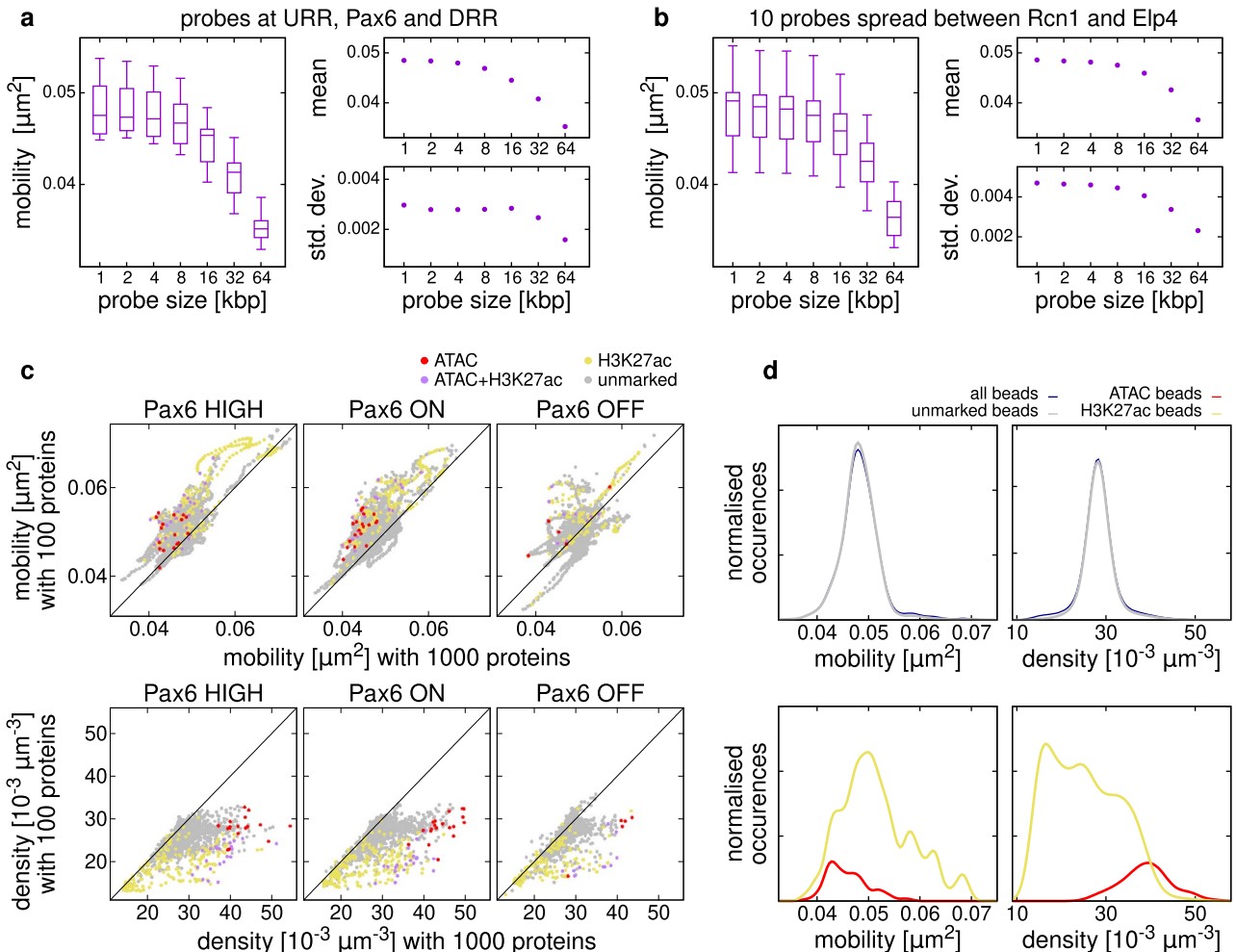

**Extended Data Fig. 4 | Factors affecting mobility. a**, Probe size. Mobility is defined as the MSD of the centre of mass of the beads covered by a probe at a given lag time of $10^4\tau \approx 20.7$ s. Left: box-plots show mobilities of probes of different sizes centred at the URR, *Pax6*, and DRR (same centre positions as FISH probes detailed in Supplementary Table 2; data from all three cell lines are included, same simulations as Fig. 2). Right: both the mean and standard deviation of the distribution decreases as probe size increases. **b**, Similar plots are shown but for a set of 10 probes positioned at equally spaced points between *Rcn1* and *Elp4*. **c**, Model parameters affect mobility and density. Scatter plots compare quantities measured from simulations with 1000 proteins and with 100 proteins. Each point represents a single chromatin bead; color indicates bead properties as shown. Top: mobilities in the two cases; points above the diagonal indicate a higher mobility in simulations with fewer proteins. Bottom: densities around each bead; points below the diagonal indicate a lower density in simulations with fewer proteins. Values obtained from averages over 400

configurations of each of 20 locus copies for the 1000 protein case, and over 200 configurations of each of 10 locus copies for the 100 proteins case. **d**, Local chromatin context affects mobility and density. Histograms show mobility and density for beads across all cell lines (1000 protein simulations as in Fig. 2). Values are split according to the properties of the bead as indicated. Top-left: unmarked beads show the same mobility distribution as all beads. Bottom-left: although generally ATAC beads have lower mobility and H3K27ac beads higher mobility, both can take a broad distribution of values. Density distributions (right) show similar but opposite trends. Note that just over half of the ATAC beads are also marked with H3K27ac, so it is expected that these distributions overlap. Curves are kernel density plots obtained by summing a Gaussian function for each bead; Gaussians have widths 0.001 $\mu m^2$ for mobility and $2 \times 10^{-3} \mu m^{-3}$ for density. Curves are scaled so that they enclose and area equal to the number of beads represented.

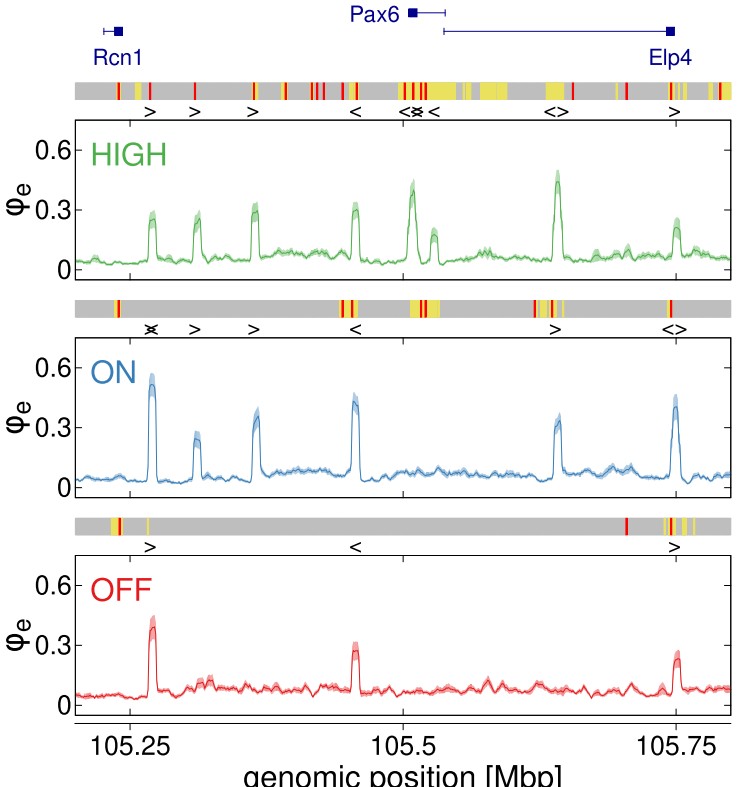

**Extended Data Fig. 5 | Extrusion probability.** Plots showing the probability that a loop extruder is positioned at a chromatin bead at any point during the simulation. The dark lines show the average over 20 copies of the locus, with the shaded region (which is often covered by the line) showing the s.e.m. The colored block above each profile shows positions of ATAC (red) and H3K27ac (yellow) beads, and the chevrons indicate the positions and orientation of CTCF sites. Recall that in each copy of the locus the CTCF sites are occupied with probability determined by the peak height in the CTCF ChIP data (see Supplementary Notes).

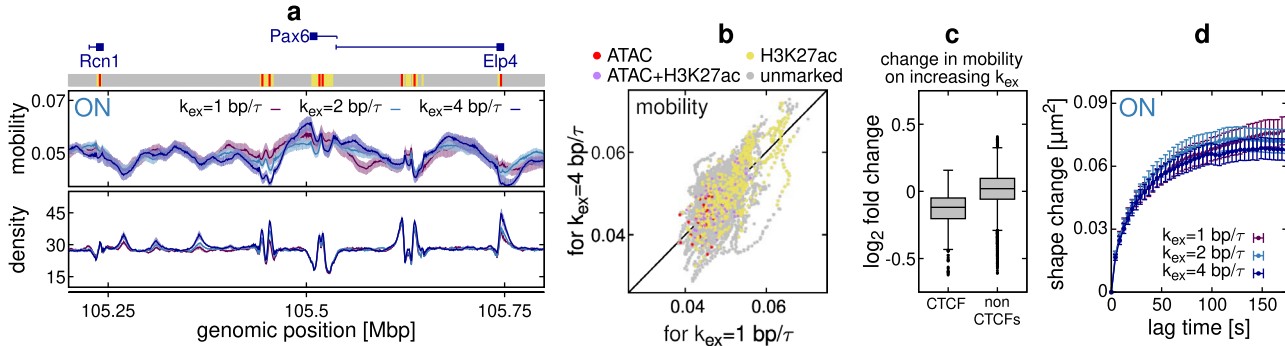

**Extended Data Fig. 6 | Effect of varying extrusion rate. a**, Plot showing the chromatin mobility (units ms$^2$) and mean local density (units $10^{-3}$ kbp $\mu$m$^{-2}$) profiles across the locus in *Pax6* ON cells for different values of the extrusion rate $k_{ex}$. The lines indicate values for single polymer beads (each representing 1 kbp of chromatin); the shaded regions indicate the s.e.m. (for the density this is typically similar in size to the line width). The colored block above the plot shows the input data used (yellow are H3K27ac regions, red are binding sites inferred from ATAC-seq peaks and other regions are grey). **b**, Scatter plot showing the mobility of all beads for different values of $k_{ex}$. Each point represents a single polymer bead, and data from the whole 3 Mbp simulated region in each of the three cell lines (*Pax6* HIGH, ON and OFF) are included. Points above the diagonal indicate an increased mobility when $k_{ex}$ is increased, while points below the diagonal indicate a decrease. Point color indicates the type of bead (as determined from

the input data); there is no discernible effect of bead type on position with respect to the diagonal. **c**, The fold change in mobility when $k_{ex}$ was increased from 1 to 4 bp/$\tau$ was calculated for each bead; the box plots show the distribution across beads for different groups. The left-hand box includes values for polymer beads within 5 kbp of a CTCF site ($n = 692$ measurements), while the right-hand box includes values for all other beads ($n = 8308$ measurements). Box extents give the interquartile range with whiskers extending by a factor of 1.5 and the centre line giving the median. There is a clear decrease in mobility on average for CTCF proximal beads, while on average there is little change for other beads. **d**, The shape-change parameter is plotted as a function of lag time for *Pax6* ON cells, for different values of $k_{ex}$. Error bars show the s.e.m. (number of independent measurements for each point is at least $n = 3570$). There is no difference between the three curves within the errors.

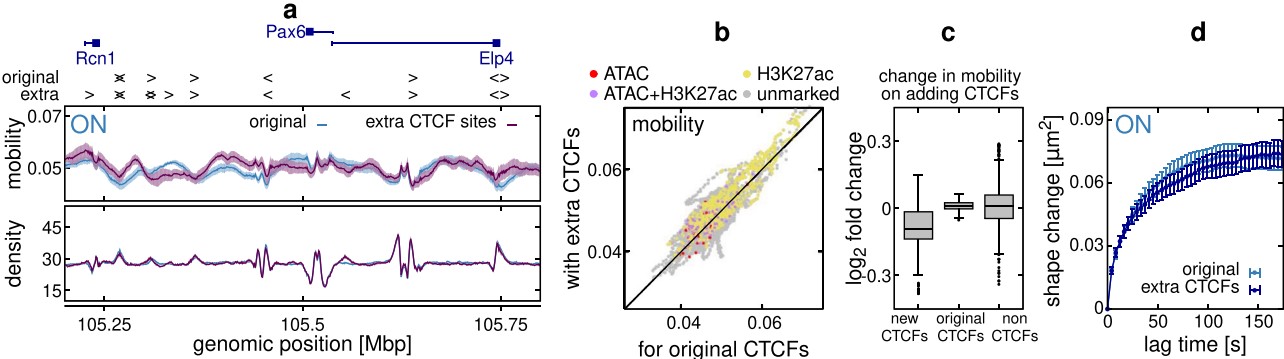

**Extended Data Fig. 7 | Effect of varying number of CTCF sites. a**, Plot showing the chromatin mobility (units ms$^2$) and mean local density (units 10$^{-3}$ kbp $\mu$m$^{-2}$) profiles across the locus in *Pax6* ON cells for simulations with and without additional CTCF sites added at random positions. Above the plot, positions and orientations of CTCF sites are shown. New CTCF sites were added at random positions across the 3 Mbp simulated region (not just within the 600 kbp region shown), such that the total number of sites was increased by 50% (see Supplementary Notes for details). The shaded regions indicate the s.e.m. (for the density this is typically similar in size to the line width). **b**, Scatter plot showing the mobility of all beads before and after the new CTCF sites were added. Point color indicates the type of bead; there is no discernible effect of bead type on position with respect to the diagonal. Data from the whole 3 Mbp simulated region in each of the three cell lines is included. **c**, The fold change in mobility when CTCF sites are added was measured for each bead, and beads split into three groups: beads within 5 kbp of newly added CTCF sites (*n* = 369 measurements), beads within 5 kbp of the original set of CTCF sites (*n* = 33 measurements), and all other beads (*n* = 7939 measurements). The box plots show the distributions of log$_2$ fold change across beads for each group. Box extents give the interquartile range with whiskers extending by a factor of 1.5 and the centre line giving the median. Mobility at new CTCF sites decreases on average, while at existing CTCF site it remains largely unchanged. At other beads, on average there is little change, but many individual beads show a small increase or decrease. **d**, The shape-change parameter is plotted as a function of lag time for *Pax6* ON cells, for simulations with and without the additional CTCF sites (error bars show s.e.m. and for each point there were at least *n* = 3570 independent measurement). There is no difference between the curves within the errors.

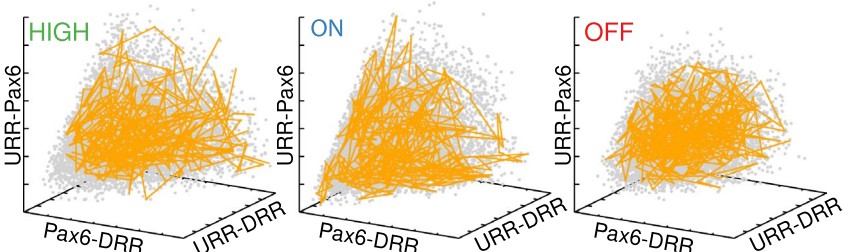

**Extended Data Fig. 8 | The *Pax6* locus explores a 'configuration space'.** Scatter plots showing the separations of the *Pax6* and enhancer probes in each of the different cell lines, similar to Fig. 4c. Each grey point represents a single time point; data are shown for all 20 repeat simulations of the locus in each case. The size of the cloud of points in this configuration space represents the variability of the locus. In each plot the yellow line shows a trajectory from a single representative simulation, that is, the path through configuration space is shown. The time interval between points along the path is equivalent to about 4 s, and the total duration of each simulation is about 27 minutes.

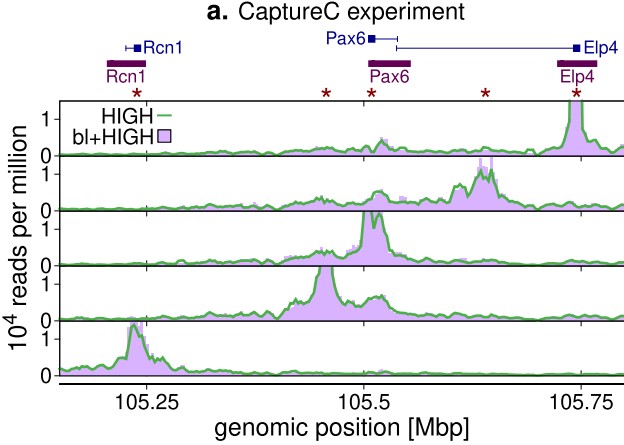

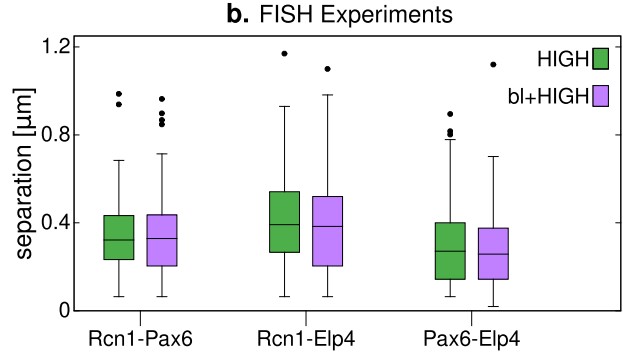

**Extended Data Fig. 9 | Bleomycin treatment relieves superhelical tension.**
**a**, Plots showing CaptureC data from *Pax6* HIGH cells with and without bleomycin treatment. Data from five viewpoints are shown, with viewpoint positions indicated by stars above the plots. Positions of genes are also indicated. **b**, Single cell microscopy was used to measure separations of FISH probes positioned at *Pax6* and the two neighbouring genes *Rcn1* and *Elp4* (positions are indicated under the genes in panel **a**). Box plots show the distributions of separations for

$n = 106$ measurements for bleomycin treated cells and $n = 104$ measurements for untreated cells. Box extents give the interquartile range with whiskers extending by a factor of 1.5 and the centre line giving the median. A Mann-Whitney U test could not reject the null hypothesis that measurements are obtained from the same distribution ($P > 0.05$ for all three pairs of probes; from left to right $P = 0.98$, $P = 0.39$ and $P = 0.31$).

| | |
|---|---|

# Reporting Summary

## Statistics

For all statistical analyses, confirm that the following items are present in the figure legend, table legend, main text, or Methods section.

| n/a | Confirmed | |
|---|---|---|
| ☐ | ☒ | The exact sample size ($n$) for each experimental group/condition, given as a discrete number and unit of measurement |
| ☒ | ☐ | A statement on whether measurements were taken from distinct samples or whether the same sample was measured repeatedly |
| ☐ | ☒ | The statistical test(s) used AND whether they are one- or two-sided *Only common tests should be described solely by name; describe more complex techniques in the Methods section.* |
| ☒ | ☐ | A description of all covariates tested |
| ☒ | ☐ | A description of any assumptions or corrections, such as tests of normality and adjustment for multiple comparisons |
| ☐ | ☒ | A full description of the statistical parameters including central tendency (e.g. means) or other basic estimates (e.g. regression coefficient) AND variation (e.g. standard deviation) or associated estimates of uncertainty (e.g. confidence intervals) |
| ☐ | ☒ | For null hypothesis testing, the test statistic (e.g. $F$, $t$, $r$) with confidence intervals, effect sizes, degrees of freedom and $P$ value noted *Give P values as exact values whenever suitable.* |
| ☒ | ☐ | For Bayesian analysis, information on the choice of priors and Markov chain Monte Carlo settings |
| ☒ | ☐ | For hierarchical and complex designs, identification of the appropriate level for tests and full reporting of outcomes |
| ☐ | ☒ | Estimates of effect sizes (e.g. Cohen's $d$, Pearson's $r$), indicating how they were calculated |

*Our web collection on statistics for biologists contains articles on many of the points above.*

## Software and code

Policy information about availability of computer code

| Data collection | Simulations were performed using the LAMMPS molecular dynamics software (version 12Dec2018), and the HiP-HoP model published in Buckle et al Molecular Cell 2018.  Image capture and analysis were done using scripts written for Iola Spectrum (Scanalytics). |
|---|---|
| Data analysis | CaptureC data was analyzed using the capC-MAP 1.1.3 software (which uses Bowtie1.1.1, samtools 1.3.1 and cutadapt 1.11). Previously published ATAC-seq data were analyzed using Trim Galore v0.6.5, Bowtie2 v2.4.2, MACS 2.1.1 and the Bedtools suit. Previously published ChIP-on-chip data were analyzed using the "Ringo" bioconductor tool. The MEME suit of tools was used for CTCF motif analysis. For FISH, images were collected from at least 50 randomly selected nuclei for each experiment and then analysed using Iola scripts that calculate the distance between two probe signals. Custom scripts used for analysis of simulation data have been deposited in the Edinburgh DataShare digital repository and are accessible at DOI:10.7488/ds/7477 (https://doi.org/10.7488/ds/7477). |

For manuscripts utilizing custom algorithms or software that are central to the research but not yet described in published literature, software must be made available to editors and reviewers. We strongly encourage code deposition in a community repository (e.g. GitHub). See the Nature Portfolio guidelines for submitting code & software for further information.

## Data

Policy information about availability of data

All manuscripts must include a data availability statement. This statement should provide the following information, where applicable:
- Accession codes, unique identifiers, or web links for publicly available datasets
- A description of any restrictions on data availability
- For clinical datasets or third party data, please ensure that the statement adheres to our policy

Previously published sequencing data was obtained from the NCBI Gene Expression Omnibus (GEO) as follows: ATAC-seq data GSE119656; ChIP-on-chip data

# Field-specific reporting

Please select the one below that is the best fit for your research. If you are not sure, read the appropriate sections before making your selection.

☒ Life sciences ☐ Behavioural & social sciences ☐ Ecological, evolutionary & environmental sciences

For a reference copy of the document with all sections, see nature.com/documents/nr-reporting-summary-flat.pdf

# Life sciences study design

All studies must disclose on these points even when the disclosure is negative.

| | |
|---|---|
| Sample size | For FISH at least 50 nuclei were analysed per sample. This was number was determined from a power calculation to enable true differences to be identified. Also see Naughton et al., 2013 NSMB |
| Data exclusions | No data was excluded |
| Replication | Two biological replicated were generated for all NG CaptureC experiments. Individual replicates were found to be consistent, and for paper figures were combined. |
| Randomization | Randomization was not applicable as there was no grouping in this study. |
| Blinding | The study did not involve human individuals or animals, and no subjective evaluations were performed. Blinding was therefore not applicable in this study. |

# Reporting for specific materials, systems and methods

We require information from authors about some types of materials, experimental systems and methods used in many studies. Here, indicate whether each material, system or method listed is relevant to your study. If you are not sure if a list item applies to your research, read the appropriate section before selecting a response.

### Materials & experimental systems
| n/a | Involved in the study |
|---|---|
| ☒ | ☐ Antibodies |
| ☐ | ☒ Eukaryotic cell lines |
| ☒ | ☐ Palaeontology and archaeology |
| ☒ | ☐ Animals and other organisms |
| ☒ | ☐ Human research participants |
| ☒ | ☐ Clinical data |
| ☒ | ☐ Dual use research of concern |

### Methods
| n/a | Involved in the study |
|---|---|
| ☒ | ☐ ChIP-seq |
| ☒ | ☐ Flow cytometry |
| ☒ | ☐ MRI-based neuroimaging |

# Eukaryotic cell lines

Policy information about cell lines

| | |
|---|---|
| Cell line source(s) | b-TC3 cells (referred to as Pax6-HIGH cells in the paper) were used in this study, obtained from DSMZ  (Cat# ACC-324, RRID: CVCL_0172). They have not been authenticated but have been used in many of our previous studies  e.g. Buckle, Brackley et al., 2018 Molecular Cell |
| Authentication | None |
| Mycoplasma contamination | Cell lines are routinely tested for Mycoplasma by technical services within the Institute Genetics and Cancer; the cell line used int the study tested negative. |
| Commonly misidentified lines (See ICLAC register) | No commonly misidentified cell lines were used in this study. |