## [Peer Review File · Nature Structural & Molecular Biology]

Peer Review Information

Manuscript Title: Transcription modulates chromatin dynamics and locus configuration sampling

Corresponding author name(s): Nick Gilbert, Chris Brackley

Reviewer Comments & Decisions:

Decision Letter, initial version:

Message: 25th Mar 2022

Dear Dr. Brackley,

Thank you for submitting your manuscript "Transcription modulates chromatin dynamics and locus configuration sampling". I apologize for the unusual delay in communicating a decision on your manuscript, which was due to difficulties in obtaining the full complement of referee reports as well as several Covid-19 cases in our small editorial team within the last weeks. After carefully considering the reviewers' comments (copied below), I am afraid that we cannot offer to publish your manuscript, at least in its current form, in Nature Structural & Molecular Biology.

You will see that while the Reviewer #2 is generally positive about the findings, Referees 1 and 3 raise fundamental concerns about the strength of the conclusions that can be drawn from the simulated polymer model. Reviewer #1 finds the sims convincing but that the HiP-HoP model does not accurately reproduce the observed cHi-C profiles presented, and points to some apparent discrepancies that need to be resolved to support the proposed model. Reviewer #3 also queries the extent to which the model represents the genomic data, and requests additional data that are necessary to critically evaluate the model. Editorially, we agree it is essential that these concerns be addressed, which precludes us from considering the present manuscript further for publication in NSMB.

However, if further experimentation, analysis, and revisions allow you to address the referees concerns in full, we would be prepared to consider an appeal of our decision, on the condition that no related work is published in the interim or has been accepted in our journal. Please feel free to contact me to discuss an appeal and potential revision. Please note that, until we have the opportunity to read the revised manuscript in its entirety, we cannot promise that it will be sent back for peer review.

I am sorry we could not be more positive at this time. I hope that you find the referees' comments useful in deciding how best to proceed.

With kind regards,

Beth

Beth Moorefield, Ph.D.
Senior Editor
Nature Structural & Molecular Biology

Referee expertise:

Referee #1: Structural genomics/computational biology

Referee #2: 3D genome/txn'l regulation/physical modeling

Referee #3: 3D genome/txn'l regulation/physical modeling

Reviewers' Comments:

Reviewer #1:

Remarks to the Author:

The work by Forte and colleagues introduces an extension of the already published HiP-HoP model from the same authors. The new polymer physics-based simulations claim that it can now simulate time-resolve conformational changes at the population (bulk) and single cell scales. This is accomplished by concatenating long strings of beads containing different configurations of the same locus from 3 different cell lines. However, and to this reviewer, there is no apparent and clear differences between this model and the already published one by the same authors in *Molecular Cell*, to the exception of longer simulations and larger/more diverse regions of DNA to simulate. Although I believe the simulations and methods are solid, to me the HiP-HoP model fails to accurately reproduce the observed cHi-C profiles as observed in Supplementary Figure 1. Although the authors say in the text that the simulated profiles and the cHi-C profiles are "good predictions", it is not clear to me how good the models represent the data. No statistical measures are provided and not explanation on the apparent differences already observed by eye in Supplementary Figure 1. This is essential as all the remaining results and discussion depends on relying that the model and its parameters are able to accurately represent what may be happening in the cells. The only evidences of the plausible conformation of the genomic locus in the studied cells is the FISH and cHi-C experiments, which are not clearly represented by the HiP-HoP model. This fact, together with the limited novelty of this HiP-HoP model compared to the one published in *Molecular Cell*, precludes me to recommend the work for publication. Next, I list what I think are major limitations:

1. As mentioned above, this reviewer is not convinced that the HiP-HoP model is capable to accurately reproduce the FISH and cHi-C available datasets. This is impossible to assess with no quantitative measure of the similarities of the conformations resulting from the simulations and the experimental data, which is not provided in the manuscript. This point

is essential for the rest to be taken into consideration.

2. How reliable is that Yeast chromatin dynamics is used to assess the relationship between simulation time and real time in a Mouse chromatin setting. This is also important and never discussed nor assessed.

3. What the models give back beyond what they are input? Most of the analysis performed in the simulations by the HiP-HoP model could have been assess (even analytically) by simply thinking what the imposed parameters would do to a polymer. In other words, the authors, in my opinion, fail to extract from the models more than what they impose to them. This is important to enrich our characterization of chromatin folding and dynamics.

4. Many of the decisions on the HiP-HoP model and the analysis are not justified. For example, why keep only 400 configurations of the 800,000 possible? Why to use 900kb separating regions between locus of different cell lines? Why to have mix different cell lines in the same simulation? Why 100kb is the cut-off to assess local vs non-local interactions when TADs in mouse are larger than those 100Kb? Etc. There are many decisions that are not justified, which makes think the reader whether there are alternative models that would better work to study the dynamics of chromatin.

Reviewer #2:

Remarks to the Author:

The authors use a polymeric mode, already validated in other works, to study the effect of transcription on the conformational properties and on the dynamics of chromosomal regions in different cell lines. In my opinion, the manuscript is interesting, overall correct and well written. I have just some comments on specific parts:

1) The sentence "proteins tend to come together into clusters" (p. 2) should be clarified and it should be better specified what proteins the authors refer to. As far as I know, it is not yet clear what proteins undergo liquid-liquid phase separation (independently on the polymer, so they are the cause of the structuring of the polymer) and what cause come together as a consequence of their binding to the chromatin fiber (see e.g. Erdel & Krippe, Biophysical J. 114, 2262, 2018).

2) Merging 10 copies of the system under study can indeed fasten simulations but could also cause artifacts in the results due to the interactions between the copies. Are there elements to think that this is not the case?

3) I do not understand the sentence "the configuration could still depend on the initial condition, see Supplementary Information" (p.2), nor I find an explanation in the SI. It is quite important to understand if the results should be interpreted as stationary dynamics or not. I would anyway avoid the term "equilibrium" since loop-extrusion breaks detailed balance.

4) If the goal of Figs. 2b,c,d is to highligh subdiffusion, I would plot them in log-log scale. It would be also interesting to study what exponents the model displays.

5) Usually the term "mobility" (p. 3) is used in the context of stochastic dynamics to indicate the inverse of the friction coefficient. Here it is used with a different meaning, and this could generate some confusion.

6) I am confused about how the volume of points in conformational space are calculated (p. 6 and fig. 4d). I would also expect that the interpretation of the calculated volumes depend on the fractal dimension of the arrangement of points.

7) How does the concept of "sampling" introduced in the discussion (p.10) is related to the results of Tiana et al., Biophys. J. 110, 1234 (2016)?

8) Does the extruder has a finite probability to cross CTCF (see e.g., Hansen et al. eLife 6,e25776 (2017)) ?

9) In Supplementary Materials: R_HARM and r_0 are not "equilibrium" lengths unless in absence of other forces. I would change the phrasing.

Reviewer #3:

Remarks to the Author:

Forte et al. address the question of whether variability in chromosome conformations represents cell-to-cell differences or fluctuations over time, and moreover, how the notion of strongly interacting loci such as enhancers and promoters can be reconciled with the idea of highly dynamic chromatin. They simulate the Pax6 locus in 3 different transcriptional states and study its dynamics using their previously developed polymer model. They argue that compaction/stiffness of the fiber and protein bridging, but not loop extrusion, are the dominant factors that determine chromosome configurational dynamics in their model. They find that variation in mobility of small chromosomal segments is not constraining in this model since loci rapidly sample their configurational space. However, different biophysical properties, related to different elements of the model can regulate the frequency and duration with which genomic elements interact (again, such as enhancers and promoters). The results mostly agree with previous and new CaptureC and FISH experiments, but some questions remain.

The manuscript is clear and interesting, and the main claims are potentially important in that they are somewhat strong statements about genomic structure, biophysics, and mechanisms. However, the weakness of this manuscript is that it is not clear to what extent the simulated model is representative of reasonable models of the genome (e.g., those that reproduce FISH/CaptureC) or to what extent the findings are general to reasonable polymer models. I realize that addressing this potentially difficult, but I worry that the strength of the conclusions really depends on the degree to which one takes the basic construction of the model seriously. Perhaps the authors could consider simpler polymer models (or perhaps another model, if any, that recapitulates the CaptureC and/or FISH data) and see if the general features of the dynamics are retained. Or perhaps this argument could be discussed more clearly and directly in the discussion section. Otherwise, the conclusions drawn from the modeling might be dismissed too easily.

There are a few other points that I would also like to see addressed, listed below.

Major:

1. The model includes loop extrusion, but it is largely ignored in studying the dynamics, except for Fig 2i where extruder density is correlated with mobility. But how do extruder processivity and density alter the locus variability and shape change measure (fig 4)? Relatedly, a parameter sweep may be cumbersome here, but how do extruder dynamics (e.g. processivity) alter mobility (this issue is mentioned only in passing in the results)?

2. The simulations currently do not simultaneously recapitulate the CaptureC and FISH data from transcription inhibition experiments. The authors argue that the effects due to bridging protein switch rates and bridging protein binding density could offset each other to enrich chromatin interactions without altering FISH measurements. But this is not shown. Proof of principle simulations showing this qualitative behavior would be useful even if they don't give precise agreement with the experiments.

3. Related to both of the above, evidence has emerged in the last few years that extrusion may interact with transcription, enhancers, and promoters. Could this missing aspect of the model explain the apparent shortcomings of the simulations in Fig 6?

Minor:

1. It is claimed that 40 kb regions don't exhibit large changes in mobility while 1 kb regions do. However, the 40 kb regions are selected from a ~ 200 kb window, so I'm not sure this is surprising. What if more distant 40 kb regions are compared?

2. The correlation between mobility and localness is very subtle and appears to be non-monotonic. Why doesn't the relationship between these variables continue above $0.05 \mu\text{m}^2$?

3. It makes sense to reduce the dimensionality of the problem by only studying 3 inter-locus separation (3 FISH probes) but is this too reductive? Related to my large first comment, isn't it possible that multiple models with different dynamics (or differentially organized heteropolymer fibers within the same model) could still give good agreement for these 3 FISH probes. So, for example, if the analysis of Fig. 4 were repeated with 5 or 10 FISH probes or a different 3 FISH probes, would the results generally hold?

4. In the model, there's one bridging protein switch rate and one density for that protein. To what extent would having a variety of proteins (with distinct switch rates and densities) matter for the dynamics?

5. In the alpha-amanitin experiments, there's not much change in FISH probe separations, but it appears that the probes are located in genomic positions where chromatin interactions (CaptureC) don't change. How would transcription inhibition change the inter-probe distances if the FISH probes were placed at positions where Capture C changes?

6. Please provide the duration of the alpha-amanitin treatment in the methods

7. I believe panel 1d bottom/top left indicating insets should say bottom/top right

8. The comment at bottom of page 5 left column "beads with no properties" is difficult to understand. Also should it refer to supp fig 3 not supp fig 5?

9. Fig. 4: please provide distance labels on axes for locus variability and shape change

Author Rebuttal to Initial comments

School of Physics and Astronomy
University of Edinburgh
Peter Guthrie Tait Road
Edinburgh, EH9 3FD

Dear Editor,

Thank you for sending the reviews of our work “Transcription modulates chromatin dynamics to sample locus configurations”. We were disappointed to hear that our paper was not yet suitable for publication in *Nature Structural & Molecular Biology*. However, we do believe that all of the reviewers’ concerns can be addressed. As previously discussed by email, we would therefore like to submit a revised version of our manuscript which includes new data to account for the reviewers’ questions and comments.

In our work, we used the HiP-HoP modelling scheme to study the dynamics of chromatin configuration at the *Pax6* gene locus. HiP-HoP is a polymer physics based simulation approach which predicts chromosome 3D structures using data on DNA accessibility and histone modification. We had previously developed and validated the model to study the static properties of this locus; here we extended the scheme to treat chromatin *dynamics* for the first time. Some reviewers (reviewers 1 and 3) were concerned that in our paper we did not provide a full quantitative comparison between our simulations and our FISH and Capture-C experiments. Since we had provided a quantitative validation of HiP-HoP in our previous work, we did not include this again in our manuscript; in hindsight, we agree with the reviewers that it is important to provide this analysis to validate the new extended version of HiP-HoP used here. We have therefore added this to our revision.

The other main change to our manuscript is that we have added some additional analysis of the effect of varying the parameters for the ‘loop extrusion’ component of the model. This was suggested by one reviewer (reviewer 2), and we agree that this will be informative, as this well-known model (which is one component of HiP-HoP) has been a topic of much interest in recent years. Our new results revealed that varying the extrusion parameters has a subtle and complicated effect. This is because the observed behaviour emerges from the interplay between the multiple components of the model, and changing one component changes this interplay. We believe that with these additional results, our work will be of even higher interest to the chromatin biology community.

We have also addressed the reviewers’ other more minor comments by making relevant clarifications and adding additional details to the text and supplementary methods. Below we first provide some general comments to all reviewers, detailing the main changes in our revision and explaining how we have addressed their common concerns. We then go on to address each of the reviewers’ detailed points in turn. With our revision we provide a version of the manuscript with line numbers and major changes highlighted in blue.

We hope you agree that with these improvements to our manuscript, it is now suitable for publication, and will be of high interest to the broad readership of *NSMB*.

Yours faithfully,

The Authors

General comments to all reviewers - details of major revisions

We thank the three reviewers for their thorough reading of our manuscript. As some reviewers had some similar comments, we start by detailing the major changes to the manuscript here. We then address each of the reviewers' more detailed comments in turn.

A - Validation of model against FISH and Capture-C data

A concern of reviewers 1 and 3 was how well our simulations predict the CaptureC and FISH data. On a related note, reviewer 2 asked about how combining several copies of the gene locus into a single simulation might affect our results.

To address this, we first point out that here we have used the HiP-HoP modelling framework which we first presented in our Molecular Cell paper of 2018 [Buckle *et al*, Mol Cell (2018)]. In that paper we provided a quantitative comparison between various versions of the model and experimental data. In the current work we extended the model in a number of respects; notably, we included realistic chromatin density, meaning that local variations in density are better represented. We also combined 10 copies of the locus in each simulation, which improved the simulation efficiency allowing us to collect more results (in contrast, in our previous work each copy of the locus was simulated in isolation). These changes allowed us to make detailed *dynamics* measurements which were not previously possible. While the original version of the HiP-HoP model was validated against CaptureC and FISH data, we agree that a similar validation was lacking for the improved model used here. We have now provided this within our revision, and thank the reviewers for their comments.

In our 2018 paper, to provide quantitative comparisons with data, we defined two metrics, the K-Score and the Q-Score, which quantify respectively the level of agreement between simulations and FISH data, and between simulations and Capture-C data. For the FISH data, we consider the Kolmogorov-Smirnov (K-S) statistic, which gives a distance between two distributions (here a simulated and an experimental distribution of separations between fluorescent probes). As detailed in Buckle *et al*, (2018), we calculate a K-S statistic for every pair of FISH probes, and combine these to give the K-Score, which ranges from 0 to 1; a value of 1 means there is an exact overlap between simulations and experiments. For Capture-C data we first identify peaks in all of the interaction profiles for simulation and experiments; we then count the peaks which overlap between the two. The Q-Score is then the Jaccard index for the two sets of peaks [Buckle *et al*, Mol Cell (2018)]; it ranges from 0 to 1, with a value of 1 indicating exact overlap between the experimental and simulation sets of peaks. The plots below show K-Scores and Q-Scores for our previous study and our present study. In both cases, for context we also show a random control case, generated by selecting probe separation or peak positions at random.

From these plots we note that based on these measures, our new simulations actually perform very slightly better than our previous work. We have now included these plots as a new supplementary figure in our revised manuscript, with details of the calculations given in the supplementary materials (Supplementary Methods section 4.2 and Supplementary Figure 3).

As an additional more general validation of the HiP-HoP framework, we also note that in other recent work where HiP-HoP is applied to different cell lines [Rico *et al.*, Genome Research (2022) bioRxiv 2021.03.12.434963, and bioRxiv 2022.06.09.495447], we provided several quantitative comparisons between simulations and publicly available Hi-C data (we could not do this in the present work, as Hi-C data is not available in these cell lines).

B - New simulations on the effect of variation of loop extrusion parameter

Reviewer 3 noted that our modelling framework also provides the opportunity to study how variation of the loop extruder properties affects locus dynamics. We agree that this is an interesting question, and so have now performed two sets of additional simulations.

First, to understand how the rate of loop extrusion affects the dynamics we performed new simulations where we either increased or decreased the rate by a factor of 2. Interestingly, the effects are subtle and unpredictable. One might expect that increasing extrusion rate would increase the dynamics of chromatin loci. In fact, the mobility of chromatin beads across the locus can either go *up* or *down* in response to an increasing extrusion rate. To illustrate this we can plot the mobility as a function of position across the locus for different values of the extrusion rate k_{ex} , and we can show the mobility of each bead on a scatter plot (with the mobility found for a different k_{ex} on each axis):

We include a full discussion of these new results in our revision, but the main driver of the behaviour is that when the extruder rate is larger, the extruders are more likely to reach (and become halted at) CTCF sites, forming longer lived loops at these sites. There is a reduction in mobility at these sites due to this additional constraint. Extruder mediated loops are also likely to facilitate/stabilise protein mediated binding and bridging; so a shift of extruders towards CTCF sites will lead to a redistribution of protein binding across the locus, changing the pattern of mobility.

As a second method to better understand how loop extrusion affects the dynamics, we also performed a set of simulations where we increased the density of CTCF sites by 50%. We added new sites at random positions through the locus. Again, we found subtle and surprisingly nontrivial results: the mobility increases in some regions and decreases in others. The largest difference is at the new CTCF sites themselves, which show an increased local density and decreased mobility, consistent with them being involved in halted extruder mediated looping. Other small changes in mobility again likely arise due to a redistribution of protein binding, driven by looping at the new CTCF sites:

Since the effects are relatively small, and the conclusions are an aside to our main focus, we present these results in a new supplementary figure (Supplementary Figure 6), along with two new paragraphs in the main text (lines 299-319 and 497-508) and some further details in the supplement (Supplementary Methods section 5).

Point by point response to each reviewer

We now provide a detailed response to each reviewer in turn. The reviewer's comments are given in italic blue text, with our response in larger black text.

Reviewer #1:

The work by Forte and colleagues introduces an extension of the already published HiP-HoP model from the same authors. The new polymer physics-based simulations claim that it can now simulate time-resolve conformational changes at the population (bulk) and single cell scales. This is accomplished by concatenating long strings of beads containing different configurations of the same locus from 3 different cell lines. However, and to this reviewer, there is no apparent and clear differences between this model and the already published one by the same authors in Molecular Cell, to the exception of longer simulations and larger/more diverse regions of DNA to simulate. Although I believe the simulations and methods are solid, to me the HiP-HoP model fails to accurately reproduce the observed cHi-C profiles as observed in Supplementary Figure 1. Although the authors say in the text that the simulated profiles and the cHi-C profiles are "good predictions", it is not clear to me how good the models represent the data. No statistical measures are provided and not explanation on the apparent differences already observed by eye in Supplementary Figure 1. This is essential as all the remaining results and discussion depends on relying that the model and its parameters are able to accurately represent what may be happening in the cells. The only evidences of the plausible conformation of the genomic locus in the studied cells is the FISH and cHi-C experiments, which are not clearly represented by the HiP-HoP model. This fact, together with the limited novelty of this HiP-HoP model compared to the one published in Molecular Cell, precludes me to recommend the work for publication. Next, I list what I think are major limitations:

We would first like to thank the reviewer for the time taken in thoroughly reviewing our manuscript. We do, though, disagree with their assessment of the novelty of our work. As the reviewer notes, our previous paper in Molecular Cell provided details of the development and validation of the HiP-HoP model. In that paper we focused on the **static** properties of the configuration of the *Pax6* gene locus. In other words, we treated the simulation configurations as "*in silico* fixed cells". In the present work, we have made several improvements to the scheme which, crucially, now allow us to extract **dynamical** information from the simulations. This is, to our knowledge, the first time that a predictive polymer model which incorporates several mechanisms has been applied to simulate chromatin dynamics at the gene locus scale. In this important respect, the work is novel. And, given recent advances in live-cell imaging, it is timely: our results have important consequences for the interpretation of future experimental work.

1. As mentioned above, this reviewer is not convinced that the HiP-HoP model is capable to accurately reproduce the FISH and cHi-C available datasets. This is impossible to assess with no quantitative measure of the similarities of the conformations resulting from the simulations and the experimental data, which is not provided in the manuscript. This point is essential for the rest to be taken into consideration.

As detailed above in **general comments to all reviewers point A**, in our revised manuscript we have now added some additional quantitative comparison between the FISH and Capture-C data sets. We believe that this shows that our model does provide good predictions of structures in fixed cells, and so its predictions on dynamical properties are highly useful.

2. How reliable is that Yeast chromatin dynamics is used to assess the relationship between simulation time and real time in a Mouse chromatin setting. This is also important and never discussed nor assessed.

First, we note that the mapping between simulation and real times is the same across all of our simulations, so this does not affect comparisons between different simulation results. It only affects comparisons to real times. We used the yeast data from Hajjoul *et al.*, Genome Research (2013), as this has been used to map simulation time units to real times in many other simulation studies. It provides MSD measurements for 7 different chromosome loci covering lag times between 0.02s and 400s, and reveals an order of magnitude estimate for the mean effective diffusion constant of $0.01 \mu\text{m}^2 \text{s}^{-0.5}$. We wanted a long time-scale mean value to do the mapping, as all of the short-time local variation which we discuss in the paper should be averaged out. We agree that it would be better to use a more recent data set from mammalian cells, however we have not found a single paper with as comprehensive a range of measurements. Many recent studies of chromatin dynamics rely on very short time scale measurements (much less than 1 second), which do not give reliable extrapolations to longer time MSDs, and many papers where MSDs are shown do not provide their data. We have nevertheless found several papers showing MSDs of chromatin loci in mammalian cells which are overall consistent with the yeast measurements, as illustrated in the following plots.

A range of effective diffusion constants and anomalous diffusion exponents have been reported for different loci in different cell types and organisms (mouse and human). As can be seen in the plots, the values for yeast are consistent with this range. It is informative to examine how using different mapping would affect our results. For example, one of our key results is that the locus explores all of its configurations within the simulation duration, which maps to about 28 minutes using the yeast data. Using data from HeLa cells from Shinkai *et al.*, PLoS

Comput Biol (2016) would give a mapping of 16 minutes for the simulation duration; using a human breast cancer cell line data from Chang *et al.*, Protein & Cell (2022) for an A-compartment locus showing an MSD in the middle of the range gives a mapping of 89 minutes; and using Hep1-6 (mouse) cell data from Duan *et al.*, Genome Biology (2018) gives a mapping of 25 minutes. We believe that this discussion would be informative for readers and thank the reviewer for raising this point; we have added a comment in the main text (lines 414-416) and some further details in the supplementary material (Supplementary Methods section 1.5).

3. What the models give back beyond what they are input? Most of the analysis performed in the simulations by the HiP-HoP model could have been assess (even analytically) by simply thinking what the imposed parameters would do to a polymer. In other words, the authors, in my opinion, fail to extract from the models more than what they impose to them. This is important to enrich our characterization of chromatin folding and dynamics.

We disagree with the reviewer that our model predictions could be inferred or “guessed” from our input. By careful analysis of our simulation trajectories, we have been able to provide rational explanations for our observations and predictions, but this would not have been possible without performing the modelling work.

From the literature it is known that polymer dynamics can be analytically studied only in simple cases (e.g., as in the Rouse model for an ideal chain, or the Zimm model for an ideal chain subject to hydrodynamic interactions). The crowding and complexity of the nuclear environment prevents the formulation of an analytical theory describing chromatin dynamics *in vivo*. On the other hand, our simulations incorporate phenomena such as dynamic TF binding and loop extrusion. Our simulations suggest that the variation in dynamics which can be observed at different loci strongly depends on probe size and position (Fig. 1a-e), casting some previous, seemingly contradictory, experimental observations in a new light. While some general trends in behaviour follow from the ‘input properties’ of different chromatin beads (e.g. less compact H3K27ac regions are more mobile), the fact that a wide range of values of measured quantities are observed for beads with the same properties reveals that the whole local pattern of chromatin modifications/accessibility/CTCF binding plays a role. Our conclusions could not have been reached without our detailed analysis of the simulations.

4. Many of the decisions on the HiP-HoP model and the analysis are not justified. For example, why keep only 400 configurations of the 800,000 possible? Why to use 900kb separating regions between locus of different cell lines? Why to have mix different cell lines in the same simulation? Why 100kb is the cut-off to assess local vs non-local interactions when TADs in mouse are larger than those 100Kb? Etc. There are many decisions that are not justified, which makes think the reader whether there are alternative models that would better work to study the dynamics of chromatin.

We agree that the rationale for these various choices should be more clearly explained in the manuscript, and we have now added comments accordingly.

On the choice of how many configurations to keep: our simulations track the motion of the locus through time. Therefore, two configurations taken a short time interval apart will be highly similar. The value was chosen to balance between not sampling too quickly (saving many configurations which are practically identical), while still saving configurations regularly enough to capture the dynamics. We have re-written the section of the main text on this to

clarify (lines 175-185) and have added further detail on the choice to the supplement (Supplementary Methods section 1.4).

The choice of 900kbp between locus copies was somewhat arbitrary, but chosen so as to be of similar or larger than the typical size of chromatin domains. The idea being that the copies will be sufficiently far apart such that they will not affect each other's configuration. As detailed above, we found that the simulated configurations represented the FISH and Capture-C data similarly to previous work where each copy of the locus was simulated in isolation. We have added a comment on this to the supplement (Supplementary Methods section 1.4).

We mixed the different cell lines in different simulations in order that each copy of the locus was embedded within a similar "background" environment. Particularly, the same number of model proteins was used in each simulation. Different cell lines have different numbers of protein binding sites; by including a mixture of cell lines in each simulation, the overall ratio between proteins and binding sites was always approximately constant. We have added a sentence to the supplement to clarify this (Supplementary Methods section 1.4).

The 100kbp cut-off was chosen as a threshold since it is slightly smaller than the expected mean promoter-enhancer loop size, so that cis-regulatory loops can be counted as long-ranged. This is therefore relevant for *Pax6*, but a different threshold may be more informative for other loci. We have added a sentence clarifying this in the paper main text (lines 348-349)

The other model parameters were mainly chosen based on our previous work. Since HiP-HoP simulations are computationally intensive, a systematic parameter optimization sweep is not possible. Instead, parameters were chosen based on reasonable estimates, and the effect of varying these determined in test simulations (see [Buckle *et al*, Mol Cell (2018)]). We have now stated this more clearly in the text (Supplementary Methods section 1.5).

Reviewer #2:

The authors use a polymeric mode, already validated in other works, to study the effect of transcription on the conformational properties and on the dynamics of chromosomal regions in different cell lines. In my opinion, the manuscript is interesting, overall correct and well written. I have just some comments on specific parts:

We thank the reviewer for their thorough review of our work. We are glad that they found the manuscript to be interesting, correct and well written. We now address each of their specific comments in turn.

1) The sentence "proteins tend to come together into clusters" (p. 2) should be clarified and it should be better specified what proteins the authors refer to. As far as I know, it is not yet clear what proteins undergo liquid-liquid phase separation (independently on the polymer, so they are the cause of the structuring of the polymer) and what cause come together as a consequence of their binding to the chromatin fiber (see e.g. Erdel & Krippel, Biophysical J. 114, 2262, 2018).

We agree that this was unclear in our original manuscript. Here we were talking about the model proteins in our simulations. Within polymer simulations, any proteins which can form molecular bridges between distant segments of a polymer will tend to come together via a “bridging-induced attraction”. We first demonstrated this in simulations some years ago [Brackley *et al*, PNAS (2013)] and it has more recently been demonstrated *in vitro* [Ryu *et al*, Science Advances (2021)]. The point we wanted to make here is that the clustering observed in our simulations is reminiscent of recent observations of foci of various transcription-related proteins (e.g. [Chong *et al*, Science (2018)] or [Shrinivas *et al*, Mol Cell (2019)]). While in our model the clustering is driven by protein-chromatin interactions, as the reviewer notes, protein-protein interactions and liquid phase separation may well play an important role. Indeed, it would be interesting to study the interplay between these effects in the future. We have now clarified this section in our revision (lines 117-118 and 124-125).

2) Merging 10 copies of the system under study can indeed fasten simulations but could also cause artifacts in the results due to the interactions between the copies. Are there elements to think that this is not the case?

As detailed in **general comments to all reviewers point A** above, we have now provided some additional quantitative comparison between our simulations and experimental data (FISH and CaptureC), and between our simulations and previous work where a single copy of the *Pax6* locus was simulated in isolation. We see similar agreement between data and both sets of simulations, which suggests that (while they can and do interact) the presence of other copies of the locus in the simulations does not have a strong effect on the configurations. We can of course not be completely sure that the broader surroundings do not have an effect, and this is a limitation of the model. Instead of merging 10 copies, one could instead run 10 independent simulations of a 40Mbp region around *Pax6*, but this would be prohibitively computationally expensive, and our analysis suggests that there would be little change to our results.

3) I do not understand the sentence "the configuration could still depend on the initial condition, see Supplementary Information" (p.2), nor I find an explanation in the SI. It is quite important to understand if the results should be interpreted as stationary dynamics or not. I would anyway avoid the term "equilibrium" since loop-extrusion breaks detailed balance.

We agree that this section could have been clearer. In our simulations we initialise the system in a configuration reminiscent of a mitotic chromosome (similar to the approach in [Rosa and Everaers, PLOS Comput Biol (2008)]). We then run the simulations for sufficiently long before starting to take measurements, such that the system will have reached a configuration consistent with interphase *in vivo*. As the reviewer notes, we believe that we have reached a steady state in terms of the dynamics. We have now made this clearer in the text and have added some detail to the supplement (lines 175-185 and Supplementary Methods section 1.4). We agree with the reviewer that the term “equilibrium” is not helpful in this context, and so now avoid it in the text.

4) If the goal of Figs. 2b,c,d is to highlight subdiffusion, I would plot them in log-log scale. It would be also interesting to study what exponents the model displays.

We agree that it is useful to show as a log-scale. The linear scale may also be informative, so we now include both as an inset (Figs 2b,c).

5) Usually the term "mobility" (p. 3) is used in the context of stochastic dynamics to indicate the inverse of the friction coefficient. Here it is used with a different meaning, and this could generate some confusion.

We have now added a sentence in the introduction section to clarify our meaning of the term mobility (lines 72-74). We have also stated the definition of the mobility measure more clearly when it is first introduced (lines 241-242). We believe that this should now not cause any confusion.

6) I am confused about how the volume of points in conformational space are calculated (p. 6 and fig. 4d). I would also expect that the interpretation of the calculated volumes depend on the fractal dimension of the arrangement of points.

Here we make a very simple estimate of the volume by calculating the radius of gyration of the cloud of points (detailed in Supplementary Methods section 4.7). We agree that this is only very loosely a measure of 'volume', and will depend on the spatial arrangement (or fractal dimension) of the points within the cloud. From the plots, the points can be seen to fill the volume roughly uniformly, and so we believe that this crude measure is informative. We have now added some clarification to the supplementary methods (Supplementary Methods section 4.7).

7) How does the concept of "sampling" introduced in the discussion (p.10) is related to the results of Tiana et al., Biophys. J. 110, 1234 (2016)?

The paper which the reviewer mentions uses a different type of polymer modelling approach to generate chromatin configurations, and they focus on configuration at the topologically associated domain (TAD) scale. That work predicted that TAD structures would change dynamically on a time scale shorter than the cell cycle, and that there are not large energy barriers between different configurational states. We look more specifically at the gene locus scale, and our results are consistent and complementary. The main difference in the approaches is that the Tiana *et al.* paper uses a model with an effective polymer interaction potential which is obtained by fitting to/optimised against 5C data. In our approach we instead explicitly model the mechanisms which drive the configurations. In this sense we believe that our work provides a significant advance in terms of understanding and mechanistic insight. We have now added a comment to the discussion and reference to this paper (lines 732-736).

8) Does the extruder has a finite probability to cross CTCF (see e.g., Hansen et al. eLife 6,e25776 (2017)) ?

In our model the extruders cannot cross the CTCF sites. However, across the set of copies of the locus, the CTCF sites are included stochastically with a probability based on the ChIP-seq peak height in the input data. That is to say, a given CTCF site is present in only a subset of locus copies/repeat simulations. We believe that at a population average level, this will have the same effect as allowing extruders to cross CTCFs with a finite probability. We agree that, for a given locus copy/simulation run, this might have an effect. The Hansen paper suggests a mean CTCF residence time at binding sites of between 1 and 4 min, though they note that this likely varies broadly between different sites. In our model the CTCF residence time is effectively the duration of the simulation (~50 min); however, the extruder behaviour is largely dependent on the extrusion rate and the extruder unbinding rate (the mean time an extruder remains halted at a CTCF site will be significantly shorter than the extruder unbinding time, roughly 1.3 min). It is therefore very unlikely that this will have any significant effect on our results. We have now added a comment on this to the supplement (Supplementary Methods section 2).

As detailed in **general comments to all reviewers point B** above, we have now added some additional simulations where we vary the extruder properties, and our implementation of extruders and CTCF site may be more important for cases where the extruder rate is large and there are additional CTCFs. A fuller study of the fine details of loop extruders would be interesting, but is beyond the scope of the present work.

9) In Supplementary Materials: R_{HARM} and r_0 are not "equilibrium" lengths unless in absence of other forces. I would change the phrasing.

We have now rephrased to remove the word "equilibrium" (Supplementary Methods sections 1.1 and 1.3).

Reviewer #3:

Forte et al. address the question of whether variability in chromosome conformations represents cell-to-cell differences or fluctuations over time, and moreover, how the notion of strongly interacting loci such as enhancers and promoters can be reconciled with the idea of highly dynamic chromatin. They simulate the Pax6 locus in 3 different transcriptional states and study its dynamics using their previously developed polymer model. They argue that compaction/stiffness of the fiber and protein bridging, but not loop extrusion, are the dominant factors that determine chromosome configurational dynamics in their model. They find that variation in mobility of small chromosomal segments is not constraining in this model since loci rapidly sample their configurational space. However, different biophysical properties, related to different elements of the model can regulate the frequency and duration with which genomic elements interact (again, such as enhancers and promoters). The results mostly agree with previous and new CaptureC and FISH experiments, but some questions remain.

The manuscript is clear and interesting, and the main claims are potentially important in that they are somewhat strong statements about genomic structure, biophysics, and mechanisms.

We thank the reviewer for their time and thorough review of our work. We are glad that they found our paper clear and interesting, and our results important. We now address each of their concerns in turn.

However, the weakness of this manuscript is that it is not clear to what extent the simulated model is representative of reasonable models of the genome (e.g., those that reproduce FISH/CaptureC) or to what extent the findings are general to reasonable polymer models. I realize that addressing this potentially difficult, but I worry that the strength of the conclusions really depends on the degree to which one takes the basic construction of the model seriously. Perhaps the authors could consider simpler polymer models (or perhaps another model, if any, that recapitulates the CaptureC and/or FISH data) and see if the general features of the dynamics are retained. Or perhaps this argument could be discussed more clearly and directly in the discussion section. Otherwise, the conclusions drawn from the modeling might be dismissed too easily.

The reviewer asks whether we expect our results to be specific to the HiP-HoP model, or if we would expect similar observations in other, simpler models. Before developing HiP-HoP, our group worked with several different polymer models, e.g., with different resolutions, including different species of protein, and with and without loop extrusion [Brackley *et al*, Nucleic Acids Res (2016); Brackley *et al*, Genome Biology (2016); Buckle *et al*, Mol Cell (2018)]. Given our experience with that work, we would expect the general timescale of changes to the locus configuration (e.g., as measured by the shape change parameter in Fig. 4) to be largely similar in all of these simpler polymer models. However, features such as the variation in mobility across the locus (e.g., Fig. 2e), which we have shown to depend on local heteromorphic polymer structure and CTCF position etc., would clearly not appear in a simpler model which does not have those “ingredients”. In our 2018 Molecular Cell paper, we showed that all of the HiP-HoP model ingredients were required in order to obtain good predictions of the data at *Pax6*. We therefore do not believe it is within the scope of the present work to study simpler models which we know will not give good predictions. We also point out that many of the other approaches used to model 3D chromosome structure are based on fitting or machine learning from existing Hi-C data, and by their nature most do not provide meaningful information on chromatin *dynamics* at the gene locus scale. Nevertheless, we agree it is worth adding some comment on other models to the discussion, as the reviewer suggests, and have added this to the discussion section (lines 766-773).

There are a few other points that I would also like to see addressed, listed below.

Major:

1. The model includes loop extrusion, but it is largely ignored in studying the dynamics, except for Fig 2i where extruder density is correlated with mobility. But how do extruder processivity and density alter the locus variability and shape change measure (fig 4)? Relatedly, a parameter sweep may be cumbersome here, but how do extruder dynamics (e.g. processivity) alter mobility (this issue is mentioned only in passing in the results)?

As detailed above in **general comments to all reviewers point B**, we have now performed some additional simulations and analysis studying the effect of changing the loop extruder

parameters. We believe that this addition has strengthened our work, and we thank the reviewer for their suggestion.

2. The simulations currently do not simultaneously recapitulate the CaptureC and FISH data from transcription inhibition experiments. The authors argue that the effects due to bridging protein switch rates and bridging protein binding density could offset each other to enrich chromatin interactions without altering FISH measurements. But this is not. Proof of principle simulations showing this qualitative behavior would be useful even if they don't give precise agreement with the experiments.

As the reviewer notes, we suggested that a possible explanation for the alpha amanitin treatment observations is that some protein-chromatin bridges become stabilised, leading to increased local CaptureC interactions, while binding/bridging at other sites becomes abrogated. This was a speculation, and our main message is that the effect of this treatment is complex and not well understood. We simulated the two simplest possible mechanisms one might think of, and showed that neither gives a sufficient prediction of the experimental observations. We feel that simulating a more complicated scenario would involve, at this stage, too much speculation and too many unknown quantities to provide any further insight. Since our model does not directly include transcription, a further study would be beyond the scope of this work. We have added a comment on this to the text (lines 631-634)

3. Related to both of the above, evidence has emerged in the last few years that extrusion may interact with transcription, enhancers, and promoters. Could this missing aspect of the model explain the apparent shortcomings of the simulations in Fig 6?

This is an interesting idea. It may well be that inhibiting transcription alters the loop extruder dynamics in a way that affects the locus structure. One might test this experimentally, for example, by comparing Hi-C data before and after transcription inhibition to see if features of the interaction map associated with loop extrusion show any change. On the other hand, experiments where cohesin is degraded tend not to show large changes in transcription. The link between extrusion and transcription remains unclear. Again, to address this within a simulation will likely require a model which more directly accounts for transcription than is currently possible within the HiP-HoP framework.

Minor:

1. It is claimed that 40 kb regions don't exhibit large changes in mobility while 1 kb regions do. However, the 40 kb regions are selected from a ~200 kb window, so I'm not sure this is surprising. What if more distant 40 kb regions are compared?

We agree with the reviewer that perhaps it is not surprising that the dynamics of two 40kb regions separated by 200kbp are similar. Our other results are more striking; for example, (a) when comparing dynamics for different cell lines where the expression level of the gene and the underlying chromatin properties are very different, the dynamics of the 40kb probes also look very similar (Fig 2b-d, the three curves are overlapping in each case); and (b) if we consider a 1kbp probe, there is a large variation in dynamical properties across the 200kb region. Point (a) would suggest that the MSD of any 40kbp probe would look similar (at least probes within a generally active region such as the one studied here; one might expect different dynamics in e.g., a heterochromatic region).

2. The correlation between mobility and localness is very subtle and appears to be non-monotonic. Why doesn't the relationship between these variables continue above 0.05 μm^2 ?

We agree that this plot is nontrivial, and we do not have a simple explanation. We note that there are very few beads which have mobilities above 0.05 μm^2 , and so the trend at large values is based on far fewer data points (the error bar also becomes larger). For this reason, we do not think we can definitely conclude a non-monotonic relationship, and indeed the overall correlation is positive. We have added a further comment to the text (line 366).

3. It makes sense to reduce the dimensionality of the problem by only studying 3 inter-locus separation (3 FISH probes) but is this too reductive? Related to my large first comment, isn't it possible that multiple models with different dynamics (or differentially organized heteropolymer fibers within the same model) could still give good agreement for these 3 FISH probes. So, for example, if the analysis of Fig. 4 were repeated with 5 or 10 FISH probes or a different 3 FISH probes, would the results generally hold?

The reviewer makes a good point, that only studying 3 probes is a simplification. This choice was mainly motivated by the fact that we had experimental data on fixed cells with these three probes, and so it is more straightforward to see the connection with a real system. We do, however, believe that having 3 probes is particularly relevant for *Pax6*, since they cover the gene promoter and the two distal enhancers which have been previously identified. One therefore expects that the arrangements of these three regions will be functionally relevant. For a larger locus with more regulatory elements, it may be natural to consider more probes. We have added a sentence on this to the text (lines 509-516).

On the reviewer's specific point about if we were to use 5 or 10 probes in the *Pax6* locus: we could calculate the shape change function shown in Fig. 4f,g for motion in a higher dimensional space. While the system might move through this space at a different rate, the main conclusion - that the motion through configuration space is quicker for active/more open chromatin, and the locus can visit most of the accessible configurations within a short time - would be unlikely to change. For the point on what might change if we chose three different probes, again we think our general conclusions would be unchanged. In comparing the different cell lines, we showed that the precise dynamics are affected by the local chromatin properties (Fig 4f), but the overall order of magnitude of the timescale for exploration of configuration space is not (there is not a large difference between curves in Fig 4g). As already noted in the text, we would expect slower dynamics for a larger locus (i.e., if the probes were further apart).

4. In the model, there's one bridging protein switch rate and one density for that protein. To what extent would having a variety of proteins (with distinct switch rates and densities) matter for the dynamics?

We have simulated an active region of the genome and the bridging proteins correspond to complexes of transcription factors and polymerase. We agree with the reviewer in that a natural extension of the model is to include additional species of proteins. For example, in other recent work [Rico et al, Genome Research (2022) bioRxiv 2021.03.12.434963, and bioRxiv 2022.06.09.495447] we included model proteins which bind to heterochromatin (as identified by histone modification data) and model proteins which bind to regions with polycomb associated histone marks. Certainly, this simulation set-up would allow us to investigate whether active and inactive chromatin display different dynamics and would be an

interesting future study. However, it is reasonable to think that the introduction of new inactive transcription factors would not heavily affect the dynamics of *Pax6* in these cell lines where the region is generally active (even in the cell line where *Pax6* is not expressed, the surrounding genes are).

One could also consider a more detailed model with multiple species of active protein, with different densities and switching rates. Varying the rates for one protein already had subtle and complicated effects, so we would expect an even more complicated picture for more proteins. However, we do not think this would alter the general conclusions of the work.

5. In the alpha-amanitin experiments, there's not much change in FISH probe separations, but it appears that the probes are located in genomic positions where chromatin interactions (CaptureC) don't change. How would transcription inhibition change the inter-probe distances if the FISH probes were placed at positions where Capture C changes?

Across the locus the changes to the CaptureC upon alpha amanitin treatment data are very subtle. Since the FISH probes are much larger than the interacting regions which are picked up in CaptureC, and since a peak in a CaptureC profile likely still means there are contacts in only a small number of cells, we doubt that different results would be observed for different FISH probe positions.

6. Please provide the duration of the alpha-amanitin treatment in the methods

We thank the reviewer for pointing out that we missed this detail. We have now added this to the methods section (line 850).

7. I believe panel 1d bottom/top left indicating insets should say bottom/top right

We thank the reviewer for their careful reading and noticing this error. We have fixed it in the revision.

8. The comment at bottom of page 5 left column "beads with no properties" is difficult to understand. Also should it refer to supp fig 3 not supp fig 5?

We agree that this phrasing was unclear. We have reworded in the revision (lines 324-324). Again, we thank the reviewer for noticing the incorrect figure reference, and have double checked figure references throughout.

9. Fig. 4: please provide distance labels on axes for locus variability and shape change

We believe that the reviewer is referring to axes scale labels on Figs. 4b and c. We have now added these. The similar plots in Figs. 4h and k have the same axes ranges; we have noted this in the figure caption, but have not added this to the plot so as to avoid clutter.

Decision Letter, first revision:

Message:

Dear Dr. Brackley,

Thank you for submitting your revised manuscript "Transcription modulates chromatin dynamics and locus configuration sampling" (NSMB-A45646A-Z). I apologise for the very long delay in sending you the final decision on your study.

It has now been seen by the original referees and their comments are below. The reviewers find that the paper has improved in revision, and therefore we'll be happy in principle to publish it in Nature Structural & Molecular Biology, pending minor revisions to satisfy the referees' final requests and to comply with our editorial and formatting guidelines.

To facilitate our work at this stage, we would appreciate if you could send us the main text as a word file. Please make sure to copy the NSMB account (cc'ed above).

Sincerely,

Carolina

Carolina Perdigoto, PhD
Chief Editor
Nature Structural & Molecular Biology
orcid.org/0000-0002-5783-7106

Reviewer #1 (Remarks to the Author):

I appreciate the effort that the authors have made to address (although partially) my concerns as well as those by the other reviewers. The article is much clearer now and describes more precisely what I believe the authors have done. Therefore, I see this version as more suitable for a journal like NSMB.

Reviewer #2 (Remarks to the Author):

The authors have improved considerably the quality of the manuscript, answering to all my concerns. I think that the manuscript is very interesting and should be published in its present form.

Reviewer #3 (Remarks to the Author):

The revised manuscript from Forte et al. largely addresses the reviewer comments. I am generally satisfied with the revisions, but I remain concerned that the data from three FISH probes is sufficient to constrain the system enough to make reliable predictions. In some places, the authors allude to this issue, but I think this can be discussed more clearly (detailed below). However, overall, this is a well conducted simulation study that explores hypothesized properties and mechanisms of chromatin dynamics, so I think the manuscript is a good scientific contribution.

More on my main concern and two smaller issues:

1. Would it be accurate to say that any model that recapitulates the experimental results with 3 FISH probes will give the same overall results (such as mobility vs. position along the chromatin, etc.)? The authors seem to believe that at least larger-scale dynamics (40 kb) will be insensitive to smaller scale details (page 5 lines 257-261 and 327-328: local mobility obscured for measurements of larger beads and dependence of dynamics on the environment). The answer seems important since the answer to one of the main questions — cell to cell variability vs. variability over time — may depend on this issue. So, although the revision in the discussion (lines 767-773) is good, I am a little troubled that possible differences are downplayed as “subtle.” Moreover, earlier in the discussion, lines 664-666 seem to omit what was said earlier in the manuscript about the idea that the environment (in addition to the biophysical properties) being important. Clearly, based on new the simulations, CTCF and other factors (including environmental factors from genomically distant chromatin) can affect the dynamics of the 1 kb loci; I think this should be restated in the discussion.

2. The conclusions about the mild effect of transcription on locus structure should be more limited. It may be accurate for the authors to say that transcription inhibition did not strongly affect the locus that they studied. But previous work shows that chromatin structure around individual genes or populations of genes or loci (Heinz et al. Cell 2018, Olan et al. Nat Comm 2020, Zhang et al. Sci Adv 2021, Jeppsson et

al. Sci Adv 2022, Banigan et al. bioRxiv 475367, Zhang et al. bioRxiv 498738, etc.) appears to be sensitive to transcription itself, so the authors should make minor edits to limit the scope of their conclusion here.

3. It would be helpful to include a short explanation of what “open chromatin structure” due to H3K27ac in the simulations would be helpful where H3K27ac and open structure are first mentioned (rather than only in the methods). At a minimum, the authors could make a direct reference to the HiP-HoP simulations methods subsection.

Author Rebuttal, first revision:

Response to reviewers

We thank the reviewers for their comments which we believe have helped us to improve our manuscript. We include below a point-by-point response to the final minor points made by the reviewer.

The reviewer wrote:

1. Would it be accurate to say that any model that recapitulates the experimental results with 3 FISH probes will give the same overall results (such as mobility vs. position along the chromatin, etc.)? The authors seem to believe that at least larger-scale dynamics (40 kb) will be insensitive to smaller scale details (page 5 lines 257-261 and 327-328: local mobility obscured for measurements of larger beads and dependence of dynamics on the environment). The answer seems important since the answer to one of the main questions — cell to cell variability vs. variability over time — may depend on this issue. So, although the revision in the discussion (lines 767-773) is good, I am a little troubled that possible differences are downplayed as “subtle.” Moreover, earlier in the discussion, lines 664-666 seem to omit what was said earlier in the manuscript about the idea that the environment (in addition to the biophysical properties) being important. Clearly, based on new the simulations, CTCF and other factors (including environmental factors from genomically distant chromatin) can affect the dynamics of the 1 kb loci; I think this should be restated in the discussion.

Our response:

We agree that similar mechanistic polymer models are likely to give the same order of magnitude prediction for the timescale of locus rearrangement dynamics, and we state this in the discussion section (final paragraph). However, we do not agree that *any* model would predict the same mobility vs position along the chromatin profiles; as we have shown, these profiles depend on parameters such as the number of simulated proteins and the loop extrusion parameters. There are, for example, polymer simulation schemes (e.g., the PRISMR model [Bianco et al, Nat Genet 2018]) which do not model these mechanisms directly, but incorporate their effects via an ‘effective potential’ based on fitting or machine learning. While those models can generate good predictions of ‘fixed cell’ interaction measurement, clearly if a mechanism such as loop extrusion is not included in the model, then the model will not be able to provide any insight into how loop extrusion affects dynamics.

As the reviewer suggests, we now reiterate in the discussion section that the dynamics do depend on the local environment (end of second discussion paragraph) and on model parameters (third discussion paragraph). The reviewer was concerned about our description of the effects as “subtle”. We did not intend to downplay the effects by use of this term, but instead to note that the effects of varying the parameters are difficult to understand (for example, one could not ‘guess’ the outcome without performing the simulations). We have removed the word “subtle” to avoid this confusion (final paragraph of discussion).

The reviewer wrote:

2. The conclusions about the mild effect of transcription on locus structure should be more limited. It may be accurate for the authors to say that transcription inhibition did not strongly affect the locus that they studied. But previous work shows that chromatin structure around individual genes or populations of genes or loci (Heinz et al. Cell 2018, Olan et al. Nat Comm 2020, Zhang et al. Sci Adv 2021, Jeppsson et al. Sci Adv 2022, Banigan et al. bioRxiv 475367, Zhang et al. bioRxiv 498738, etc.) appears to be sensitive to transcription itself, so the authors should make minor edits to limit the scope of their conclusion here.

Our response:

We agree that our conclusion on the effect of transcription may not apply in general at all gene loci. We have reworded the section on alpha amanitin treatment to make it clear that these observations only apply to Pax6 (“Together this suggests that, at least at Pax6, inhibiting transcription, *per se*, does not greatly affect the structure” first paragraph of the section on inhibition of transcription). In the discussion section, we already highlighted that some previous work showed the same trend as our simulations, while other previous work showed the opposite trend; we have now added that this is an area where current understanding is still poor (end of fourth discussion paragraph).

The reviewer wrote:

3. It would be helpful to include a short explanation of what “open chromatin structure” due to H3K27ac in the simulations would be helpful where H3K27ac and open structure are first mentioned (rather than only in the methods). At a minimum, the authors could make a direct reference to the HiP-HoP simulations methods subsection.

Our response:

In our revision, at the point where the different fibre structures are first mentioned we give an explanation of how this is treated in the model (first paragraph of results section), and also point the reader to further details of the implementation given in the methods sections. In the following paragraph, we also give some biological rationalisation for why we use H3K27ac data to identify these “open chromatin” regions, and give an appropriate reference. We believe that this point should now be clear to readers.

Final Decision Letter:**Message** 7th Jul 2023

:

Dear Dr. Brackley,

We are now happy to accept your revised paper "Transcription modulates chromatin dynamics and locus configuration sampling" for publication as a Article in Nature Structural & Molecular Biology.

As soon as your article is published, you can generate your shareable link by entering the DOI of your article here: http://authors.springernature.com/share. Corresponding authors will also receive an automated email with the shareable link

Your paper will be published online soon after we receive proof corrections and will appear in print in the next available issue. You can find out your date of online publication by contacting the production team shortly after sending your proof corrections. Content is published online weekly on Mondays and Thursdays, and the embargo is set at 16:00 London time (GMT)/11:00 am US Eastern time (EST) on the day of publication. Now is the

time to inform your Public Relations or Press Office about your paper, as they might be interested in promoting its publication. This will allow them time to prepare an accurate and satisfactory press release. Include your manuscript tracking number (NSMB-A45646B) and our journal name, which they will need when they contact our press office.

About one week before your paper is published online, we shall be distributing a press release to news organizations worldwide, which may very well include details of your work. We are happy for your institution or funding agency to prepare its own press release, but it must mention the embargo date and Nature Structural & Molecular Biology. If you or your Press Office have any enquiries in the meantime, please contact press@nature.com.

Please note that *Nature Structural & Molecular Biology* is a Transformative Journal (TJ). Authors may publish their research with us through the traditional subscription access route or make their paper immediately open access through payment of an article-processing charge (APC). Authors will not be required to make a final decision about access to their article until it has been accepted. [Find out more about Transformative Journals](https://www.springernature.com/gp/open-research/transformative-journals)

Authors may need to take specific actions to achieve [compliance with funder and institutional open access mandates](https://www.springernature.com/gp/open-research/funding/policy-compliance-faqs). If your research is supported by a funder that requires immediate open access (e.g. according to [Plan S principles](https://www.springernature.com/gp/open-research/plan-s-compliance)) then you should select the gold OA route, and we will direct you to the compliant route where possible. For authors selecting the subscription publication route, the journal's standard licensing terms will need to be accepted, including [self-archiving policies](https://www.springernature.com/gp/open-research/policies/journal-policies). Those licensing terms will supersede any other terms

that the author or any third party may assert apply to any version of the manuscript.

Sincerely,

Dimitris Typas
Associate Editor
Nature Structural & Molecular Biology
ORCID: 0000-0002-8737-1319
